

# Understanding concurrent heatwaves from a meridional heat transport perspective

Valerio Lembo[1], Gabriele Messori[2, 3, 4], Davide Faranda[5, 6, 7], Vera Melinda Galfi[8], Rune Grand Graversen[9, 10], and Flavio Emanuele Pons[5, 11]

[1]Institute for Atmospheric Sciences and Climate, National Research Council of Italy, Rome, Italy
[2]Department of Earth Sciences, Uppsala University, Uppsala, Sweden
[3]Swedish Centre for Impacts of Climate Extremes (climes), Uppsala University, Uppsala, Sweden
[4]Department of Meteorology, Stockholm University, Stockholm, Sweden
[5]Laboratoire des Sciences du Climat et de l'Environnement, UMR 8212 CEA-CNRS-UVSQ, Université Paris-Saclay & IPSL, CEA Saclay, l'Orme des Merisiers, 91191, Gif-sur-Yvette, France
[6]London Mathematical Laboratory, 8 Margravine Gardens, London, W6 8RH, UK
[7]Laboratoire de Météorologie Dynamique/IPSL, Ecole Normale Supérieure, PSL Research University, Sorbonne Université, Ecole Polytechnique, IP Paris, CNRS, Paris, France
[8]Institute for Environmental Studies, Vrije Universiteit Amsterdam, the Netherlands
[9]Department of Physics and Technology, University of Tromsø, Tromsø, Norway
[10]Norwegian Meteorological Institute, Tromsø, Norway
[11]Science Partners, Paris, France

**Correspondence:** Valerio Lembo (v.lembo@isac.cnr.it)

**Abstract.** We investigate concurrent heatwaves across the Northern Hemisphere through the linkage between extremes in Meridional Heat Transport (MHT) and in hemispheric land surface temperature (LST). MHT is a crucial signature of the eddy planetary-scale circulation in the mid-latitudes, which can in turn favor the simultaneous occurrence of heatwaves in remote regions. We find that the conditional occurrence of extremely weak MHT and extremely warm hemispheric LSTs is significantly more frequent than other conditional occurrences, both in Summer (JJA) and in Winter (DJF). By leveraging case studies of anomalously warm hemispheric LSTs in both these seasons, we argue that the combination of extremely weak, in some cases equatorward, MHTs and warm LSTs in JJA depend on enhanced atmospheric troughs and ridges and the formation of atmospheric blocks. These result in a wavenumber 3 pattern, connected to an anomalous land-sea thermal contrast in the high latitudes. In DJF, the weak, albeit not equatorward, MHT – warm LST events are characterized by a suppression of the climatologically dominant wavenumber 2, which weakens the overall MHT. The flow is anomalously zonal across much of North America and Eurasia, advecting moist and mild air eastward into the continents. Overall, such dynamical pattern determines abnormally warm and widespread temperatures in North America, Eastern Europe, and China. The conditional occurrence of extremely weak MHTs and warm hemispheric LSTs is found to be related to between 30% and over 40% of extremely warm hemispheric LST days in both seasons.





## 1 Introduction

A growing body of literature evidences the hemispheric signature of regional heatwaves (cfr. Van Loon and Thompson (2023)) through the medium of large-scale circulation features such as Rossby waves (e.g. White et al. (2023)). In other words, heatwaves often do not occur in isolation, but rather multiple heatwaves co-occur in geographically remote regions – also referred to as spatial compounding. The interest in these events is motivated by the socio-economic implications of spatially compounding extreme events: in a highly interconnected world, their impacts can be larger than those of isolated extremes, notably for actors with correlated exposure (Mills, 2005; Tigchelaar et al., 2018; Kornhuber et al., 2019; Vogel et al., 2019; Gaupp et al., 2020). Moreover, co-occurring heatwaves are increasing at a faster pace than all heatwaves (Messori et al., 2024). An attempt to relate co-occurring warm spells to the large scale circulation was presented by Tuel and Martius (2023), evidencing that persistent weather patterns, such as blocking, and recurrent Rossby wave packets (Ali et al., 2021) are key to mediate the distribution of warm anomalies across the Northern Hemisphere midlatitudes.

Previous work also highlighted that the meridional heat transport in the Northern Hemisphere mid-latitudes exhibits an intermittent behavior (Swanson and Pierrehumbert, 1997; Messori and Czaja, 2013, 2014, 2015; Messori et al., 2017; Lembo et al., 2019), and that sporadic episodes of very strong or weak transport at hemispheric scale can be associated with an anomalous frequency of regional weather regimes (Lembo et al., 2022). Lembo et al. (2019) and Messori et al. (2017) also demonstrated that the interplay of waves at planetary and synoptic scales shapes these transport extremes, albeit with important scale differences between boreal Summer and Winter.

Rossby waves with specific zonal wavenumbers have been recognized as possible drivers of co-occurring heatwaves (Petoukhov et al., 2013; Kornhuber et al., 2017; Strigunova et al., 2022). Kornhuber et al. (2019) identifies two specific wave configurations (wave-5 and wave-7) linked to prolonged heat events, that are responsible for locked, high-amplitude meanders in the jet stream, leading to persistent hot and dry conditions. Wave resonance has been suggested as a triggering mechanism for the formation and amplification of such quasi-stationary wave patterns, especially in boreal Summer (Petoukhov et al., 2013; Kornhuber et al., 2017). The wave resonance is facilitated by underlying conditions, such as land-sea contrast, orography, sea-surface temperature anomalies. Quasi-stationary wave patterns have also been linked to other mechanisms, such as recurrent Rossby wave packets (Röthlisberger et al., 2019), and have been identified through other methodologies, such as the local wave activity (LWA; e.g. Chen et al. (2015); Huang and Nakamura (2016)), or lowpass-filtered mean meridional wind speed at 250 hPa (Röthlisberger et al., 2016a, b). The results obtained from these latter two approaches have been recently compared for the record-breaking 2021 Northwestern Pacific heatwave, and found to provide consistent results White et al. (2023).

The literature on large-scale and/or concurrent heatwaves has therefore primarily focused on dynamical drivers. Here, we undertake a process-based investigation grounded in large-scale thermodynamics of concurrent Northern-hemisphere heatwaves. This aligns with the necessity to improve our understanding of the underlying mechanisms, predictability and the role of anthropogenic climate change for such high-impact extreme phenomena (Domeisen et al., 2023; Lembo et al., 2024). More specifically, we seek to understand how concurrent heatwaves, analysed in terms of hemispheric land-surface temperature



(LST) anomalies, are linked to extremes in planetary-scale Rossby wave anomalies, through the medium of meridional heat
transport (MHT).

The paper is structured as follows: in Section 2 data and methods are described. The frequency of occurrence of MHT and
LST extremes, as well as their conditional occurrences are described in Section 3, while Section 4 presents a case-study of
the 2021 Northwestern Pacific heatwave through the lens of MHT. Section 5 generalises the case-study from Section 3 to
co-occurring heatwaves in JJA. Section 6 provides the corresponding analysis for DJF. An interpretation of what our results
imply in terms of linkages between concurrent heatwaves and the general atmospheric circulation is given in Section 7, while
a summary and key conclusions are drawn in Section 8.

## 2   Data and Methods

### 2.1   Data

We use ERA5 reanalysis data (Hersbach et al., 2020) over 1979–2022, at a 6-hourly time resolution for the computation of MHT
and at daily time resolution for LST anomalies. MHT is computed for the $30° – 60°$ latitudinal band (hereafter "mid-latitude
channel"), using three-dimensional fields of air temperature ($T$), specific humidity ($q$), geopotential height ($z$), and zonal and
meridional velocities ($u$ and $v$, respectively), with an interpolated horizontal resolution of $0.5 \times 0.5°$ over a rectilinear grid and
137 model vertical levels. We also analyse composite anomalies of the 2-dimensional fields: sea-level pressure (SLP), LST,
sea-surface temperature (SST), total precipitation and wind speed at 250 hPa (hereafter $V_{250}$, computed as $\sqrt{u_{250}^2 + v_{250}^2}$, with
250 denoting the 250 hPa pressure level).

### 2.2   Methods

#### 2.2.1   Computation of MHT and wavenumber decomposition

The derivation of the MHTs and the zonal wavenumber decomposition follows Graversen and Burtu (2016); Lembo et al.
(2019, 2022). We hereby briefly summarize their methodology.
First, at every vertical level, the total energy $E$ is computed as the sum of the kinetic energy $K = \frac{1}{2}\left(u^2 + v^2\right)$ and the moist
static energy $H$:

$$H = L_v q + c_p T + gz \tag{1}$$

where $q$ is specific humidity; $T$ is air temperature; $z$ is geopotential height; and the constants $L_v$, $c_p$ and $g$ are the latent heat
of vaporization, the specific heat capacity at constant pressure and the gravity acceleration, respectively.
The meridional transport of $E$ is computed across a given circle of latitude and vertically averaged as:

$$\oint \int_{p_s}^{0} v E \frac{dp}{g} dx = \oint \int_{p_s}^{0} v(H + K) \frac{dp}{g} dx \tag{2}$$



with $p_s$ is the surface pressure. The transport is defined as positive when directed northwards.

Following Graversen and Burtu (2016), at every timestep and for every vertical level the zonal wavenumber decomposition is performed as:

$$\hat{\mathcal{F}}_0(t,\phi) = D \sum_{i=1}^{M} \frac{1}{4} a_{i,0}^m \, a_{i,0}^E \qquad\qquad k = 0$$

$$\hat{\mathcal{F}}_k(t,\phi) = D \sum_{i=1}^{M} \frac{1}{2} (a_{i,k}^m \, a_{i,k}^E + b_{i,k}^v \, b_{i,k}^E) \qquad\qquad k = 1,\ldots,N$$

where the sum is over all $M$ hybrid model levels, $i$ is the hybrid-level index, and $a$ and $b$ are the Fourier coefficients defined as:

$$a_{i,k}^\Psi(t,\phi) = \frac{2}{D} \int \Psi_i(t,\phi,\lambda) \cos\left(\frac{k2\pi\lambda}{d}\right) d\lambda$$

$$b_{i,k}^\Psi(t,\phi) = \frac{2}{D} \int \Psi_i(t,\phi,\lambda) \sin\left(\frac{k2\pi\lambda}{d}\right) d\lambda$$

Here $\Psi$ is either the mass flux between two hybrid half levels $m = v\frac{dp}{g}$ or $E$, $dp = p_{i+1/2} - p_{i-1/2}$ is the pressure difference between hybrid half levels, $D = 2\pi R \cos(\phi)$ with $R$ being the Earth's radius, $k$ being the zonal wavenumber, and $t$, $\phi$, $\lambda$ the time, latitude and longitude, respectively, and $d = 360^\circ$. $N$, i.e. the highest considered zonal wavenumber, is taken to be 20, following Lembo et al. (2022). A correction for the $k = 0$ wavenumber, accounting for the zonal circulation, is applied following Liang et al. (2018), as described in Lembo et al. (2019). This is needed in order to subtract the intermittent instantaneous meridional and vertical mean mass transport from the computation and therefore address specifically the instantaneous heat transports. This transport is large on short timescales, but zero in a long-term mean.

### 2.2.2 Selection of extremes

The selection of MHT extremes follows Lembo et al. (2022)with a few additional details explained in Faranda et al. (2024) and references therein. It is based on the peak over threshold approach of Extreme Value Theory (EVT, III (1975); Coles et al. (2001)), after the deseasonalization and detrending of the data and the declustering of extremes using the so-called *"extremal index"*(Ferro and Segers, 2003)). The MHT extremes are defined in $1^\circ$ wide latitude bands within the mid-latitude channel. An EVT-based convergence algorithm is then used to determine suitable percentile thresholds for the selection of extremes in each latitudinal band.

Table 1 displays the percentage of days characterized by the occurrence of a MHT extreme in at least one latitudinal band within three $10^\circ$ wide latitudinal ranges. These percentage values are remarkably homogeneous across latitudes, although there is a somewhat larger variation when comparing strong and weak MHT extremes. This is possibly related to the different properties of the extreme MHT distributions, whose skewness has a more pronounced latitudinal variations for strong extremes than for weak extremes (cfr. Lembo et al. (2022)).

We compute LST extremes by spatially averaging daily anomalies with respect to the 1979–2022 across the $20^\circ$N– $70^\circ$N domain. We detrend and deseasonalize the anomalies, and consider DJF and JJA separately. Hemispheric warm and cold





**Table 1.** Percentage of days for which at least one latitude within each $10°$ wide latitudinal range displays an extreme MHT. The different columns show separately positive and negative MHT extremes in DJF and JJA. The bottom row shows the percentage of days featuring an extreme anywhere in the mid-latitude channel.

|  | DJF-neg | DJF-pos | JJA-neg | JJA-pos |
| --- | --- | --- | --- | --- |
| **30N-40N** | 4.4% | 4.5% | 4.3% | 5.1% |
| **40N-50N** | 4.3% | 4.5% | 4.8% | 5.0% |
| **50N-60N** | 4.6% | 4.3% | 4.5% | 4.6% |
| **30N-60N** | 8.9% | 9.1% | 10.1% | 10.5% |

extremes are then defined as all values beyond the 95th and 5th percentiles, yielding 202 and 198 days for JJA and DJF, respectively.

In each $10°$-wide latitudinal band between 30N and 60N, co-occurring extreme MHT and hemispheric-mean LST are defined as the set of days during which hemispheric LST extremes co-occur with an extreme MHT.

### 2.2.3 Significance tests

To assess the impact of MHT extremes on LST extremes, we compare at each grid point the distribution of key 2-D fields $LST$, $SST$, $SLP$ and $V_{250}$ during co-occurring hemispheric LST and MHT extremes (LST|MHT) with their distribution during LST extremes without concurrent MHT extremes (LST|$\overline{\text{MHT}}$). A statistical significance test is applied to reflect whether the composite mean of such quantities in coincidence of LST|MHT is random or reflects an emerging pattern related to these events. The testing procedure consists of three steps: first, grid points showing a consistent deviation from the mean in correspondence of co-occurring LST and MHT extremes are selected; secondly neighboring grid points are clustered and sufficiently large clusters are retained; the chosen statistical test is applied to the cluster-averaged observable, with a correction to manage false discovery rate due to multiple testing over clusters. The procedure is described in detail in appendix A. As an output, a mask is provided for each composite map, highlighting regions where anomalies are significantly larger or smaller than the overall distribution.

## 3 Combined occurrence of MHT and LST extremes

We first show in Table 2 counts for LST|MHT events in each $10°$ latitudinal band. The combination of weak MHT extremes and warm hemispheric LST extremes (hereafter $\text{LST}_{95}|\text{MHT}_{\downarrow}$) is preferred in both JJA and DJF, with 34 events found for DJF, and 35 for JJA. The lower third of the mid-latitude channel features a smaller number of such events than the other latitudinal bands, with all events recorded there also being detected in the 40 $°$N–50 $°$N latitudinal band.



**Table 2.** Number of days featuring LST|MHT events, for all combinations and both seasons, within different latitudinal ranges. The percentages of joint occurrences relative to the total number of MHT extreme events are given in brackets. Cases in which conditional occurrences exceed 10%, and the corresponding values in the full 30 °N – 60 °N latitudinal band, are highlighted in bold.

|  | DJF-neg-cold | DJF-neg-warm | DJF-pos-cold | DJF-pos-warm | JJA-neg-cold | JJA-neg-warm | JJA-pos-cold | JJA-pos-warm |
|---|---|---|---|---|---|---|---|---|
| **30N-40N** | 3 (1.7%) | 11 (6.2%) | 8 (4.5%) | 7 (3.9%) | 9 (5.2%) | 11 (6.3%) | 11 (5.3%) | 14 (6.7%) |
| **40N-50N** | 3 (1.7%) | **18 (10.5%)** | 10 (5.6%) | 7 (3.9%) | 10 (5.2%) | 16 (8.3%) | 9 (4.5%) | 7 (3.5%) |
| **50N-60N** | 6 (3.3%) | **21 (11.4%)** | 15 (8.8%) | 6 (3.5%) | 7 (3.8%) | **21 (11.5%)** | 11 (5.9%) | 7 (3.8%) |
| **30N-60N** | 9 (2.6%) | **34 (9.6%)** | 19 (5.2%) | 13 (3.6%) | 19 (4.6%) | **35 (8.5%)** | 21 (4.9%) | 22 (5.1%) |

The numbers displayed in Table 2 refer to single-day LST|MHT occurrences. However, the persistence of MHT extremes is mostly limited to one or a few days (cfr. Lembo et al. (2019)), while hemispheric LST extremes in many cases span over several days. For a standard heatwave definition, i.e. a stretch of more than 5 neighboring days exceeding the given threshold for extremely warm temperature definition (on Climate Change, 2007; Radinović and Ćurić, 2012; Coughlan de Perez et al., 2023), 9 out of the 35 combined weak MHT-warm LST events in JJA would be classified as belonging to a "heatwave". If the full length of these 9 heatwave events is taken into account, it amounts to a total of 89 days, representing 42.5% of all days exceeding the 95th percentile of the hemispheric mean LSTs. In DJF, this number amounts to 56 days, representing 30.4% of all hemispheric LST warm extremes, with 13 out of 34 $LST_{95}|MHT_\downarrow$ events belonging to heatwaves lasting more than 5 days. If heatwaves of any duration would be taken into account, 62.7% extremely warm days would be associated with heatwaves featuring at least one $LST_{95}|MHT_\downarrow$ event in JJA, only 47.5% in DJF. This hints at a different seasonal behavior of hemispheric heatwaves, compared to regional heatwaves (cfr. Tuel and Martius (2024)). We interpret this as the wintertime dynamics being more efficient in mixing the low-level atmosphere at the synoptic timescales. For sake of consistency between seasons, we considered daily occurrences of LST|MHT extremes associated with $LST_{95}$ events of any duration.

Given the high frequency of occurrence of $LST_{95}|MHT_\downarrow$ events, we first focus on analysing the SST, LST, SLP and $V_{250}$ composites of $MHT_\downarrow$ events. For sake of clarity, we focused on the latitudinal band associated with the largest significant regions in most fields, i.e. 50N–60N, while composite maps for 40N–50N are provided in Appendix B. As mentioned above, events in the 30N–40N latitudinal band are extending into the neighboring poleward band, and therefore the associated composites are not shown.

Concerning JJA (Figures 1a-d), SST and LST anomalies are typically non-significant, with the exception of LST anomalies along the southern coast of Alaska. SLP composites exhibit an area of significant high pressure off the coast of Northwestern America, possibly resembling a Pacific-North American positive phase (PNA+), consistently with Lembo et al. (2022). More pronounced anomalies and larger significant regions are generally found in DJF with respect to JJA. Mostly positive LST anomalies spread across northeastern Europe, China, and central North America. Accordingly, SLP anomalies highlight two deep troughs across Scandinavia-western Russia, and over central Canada. Ridges are on the other hand found over the central Pacific and over the North Atlantic. This somewhat counterintuitive match between warm LST and troughs over central Canada

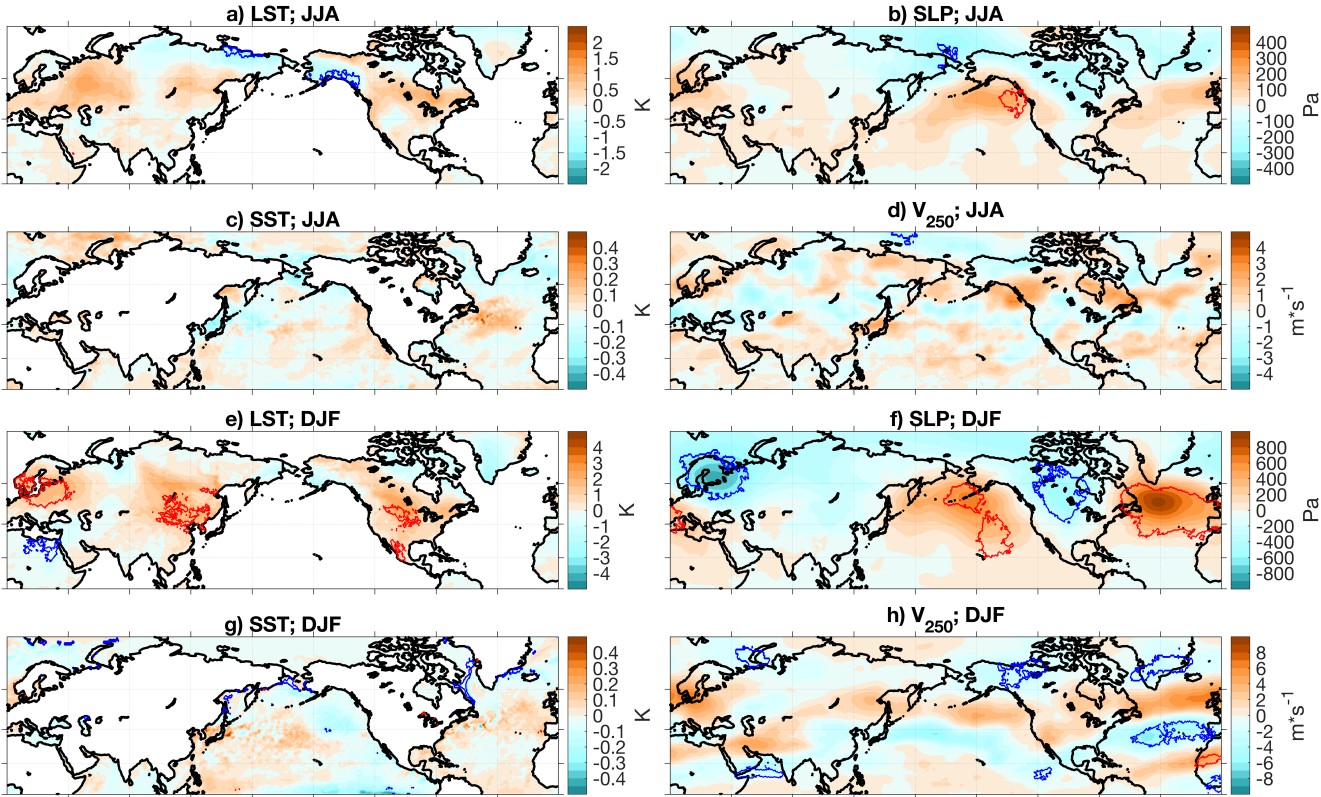

**Figure 1.** Composite mean anomaly maps for detrended-deseasonalized LST, SLP, SST and $V_{250}$ anomalies with respect to the 1979-2022 baseline period for MHT negative extremes in the 50N-60N band in JJA (a, b, c, d) and in DJF (e, f, g, h). The red/blue contours denote significant anomalies with respect to the full statistics of the field over time (see Section 2.2.3).

reflects an overall suppression of poleward/equatorward exchanges in the region, as implied by the patterns of $V_{250}$ anomalies over the Pacific. These are indeed mainly positive between 45N and 60N, suggesting that the Pacific ridge advects mild air from the ocean across the North American continent, rather than enhancing equatorward cold air advection from the Arctic. In a similar fashion as the Pacific-North American region, $V_{250}$ anomalies in the Euro-Atlantic region feature positive anomalies around 50N. In addition to that, the negative anomalies between 30N and 45N across the Atlantic and into North Africa suggest an enhanced splitting of the subpolar and subtropical jets, with a regional strengthening of the latter.

## 4 The Northwestern Pacific heatwave in 2021

In order to provide a process-based understanding of mechanisms associated with $\mathrm{LST}_{95}|\mathrm{MHT}_{\downarrow}$ events, we consider as case-study the record-breaking heatwave that affected the Pacific coastal region of North America between June and July 2021.



Previous work has elucidated the dynamics of this event. According to White et al. (2023), important elements included a strong westerly jet caused by a cold air intrusion into the Northern Pacific from East Siberia, combined with a low pressure system over Japan that enhanced a sharp temperature gradient and forced the development of a low-level moist frontal zone over an already warm and moist atmosphere in the Pacific. Successively, a south-westerly flow enhanced the formation of a very strong ridge, or blocking over the Eastern Pacific, where advected air was warmed by diabatic heating. Several of

the above mechanisms have been assessed in statistical (Pons et al., 2024) and numerical simulations (e.g. McKinnon and Simpson (2022); Mass et al. (2024)), including analyses performing Lagrangian backtracking of the air parcels responsible for the heatwave (Papritz and Röthlisberger, 2023; Baier et al., 2023). Other work has evidenced the role of topography (Lucarini et al., 2023), of Rossby waves in the Pacific excited by diabatic heating, and of the quasi-resonant amplification of planetary waves through land-surface feedbacks (cfr. Chen et al. (2023) and references therein), while low soil moisture likely had a

modest effect (although the debate is open on this topic; cfr. Osman et al. (2022); Conrick and Mass (2023); Li et al. (2024)).

Here, we provide a hemispheric context for the 2021 Northwestern Pacific heatwave, and link it to weak MHT extremes. To address this linkage, hemispheric composite anomalies averaged over 24th June – 5th July 2021 are shown in Figure 2. As expected, the largest LST anomalies (up to 10K warmer) are found in western Canada, but most of the Pacific coastal regions of North America, from California to Alaska, feature large positive anomalies (Figure 2a). Other warm anomalies are

prominent over Eastern Siberia, Central Asia, Scandinavia and Northern Africa, evidencing how this heatwave was part of a hemispheric pattern of anomalously warm temperatures. At the same time, a band of lower than average anomalies extends across Siberia into Northeastern Asia. SLP anomalies evidence that the two weeks under consideration are characterized by strong blocking over the Northeastern Pacific into Alaska, a low pressure area over the Northwestern Pacific, a Scandinavian blocking pattern and a NAO-/AO- pattern over the mid-latitude Northern Atlantic (Figure 2b). The SST anomalies over the

Northern Pacific have a noisy pattern, with strong positive anomalies in the open ocean in front of California (including the area where the frontal zone described by White et al. (2023) developed) (Figure 2c). The Northern Atlantic displays overall weak positive anomalies, particularly off the eastern coast of North America. The cold air intrusion from Eastern Siberia has its signature in the band of precipitation anomalies making its way deep into the Central Pacific (Figure 2d). Finally, the strong blocking over Alaska and the Northeastern Pacific, as well as over Scandinavia, is associated with a northward shift of the $V_{250}$

positive anomaly towards Northern Canada and the Barents Sea. On the contrary, the positive anomaly is southward-displaced towards the central Pacific and Atlantic, consistently with the SLP anomalies (Figure 2e). This suggests a displacement of both branches of the jet stream in the Northern Hemisphere.

The role of the weak MHT extreme is elucidated in Figure 3. The evolution of MHT as a function of latitudes (Figure 3a) evidences that the beginning of the heatwave in the Northwest Pacific coastal region (on 26th June 2021) coincides with

an extremely weak MHT anomaly north of 45N. In the following days, the weak transport extends to lower latitudes, and it eventually returns to climatological values by the end of June. North of 50N, this MHT event ranks second among all weak MHT extremes in the $\mathrm{LST}_{95}|\mathrm{MHT}_\downarrow$ sample (Figure 3b). It specifically displays negative transports at all latitudes north of 43N, reaching a minumum at 60N with a negative value implying overall equatorward transport. The wavenumber signature of this extremely weak transport is shown in Figure 3c. Negative MHTs are found for $k = 2$ and $k = 3$ contributions, especially



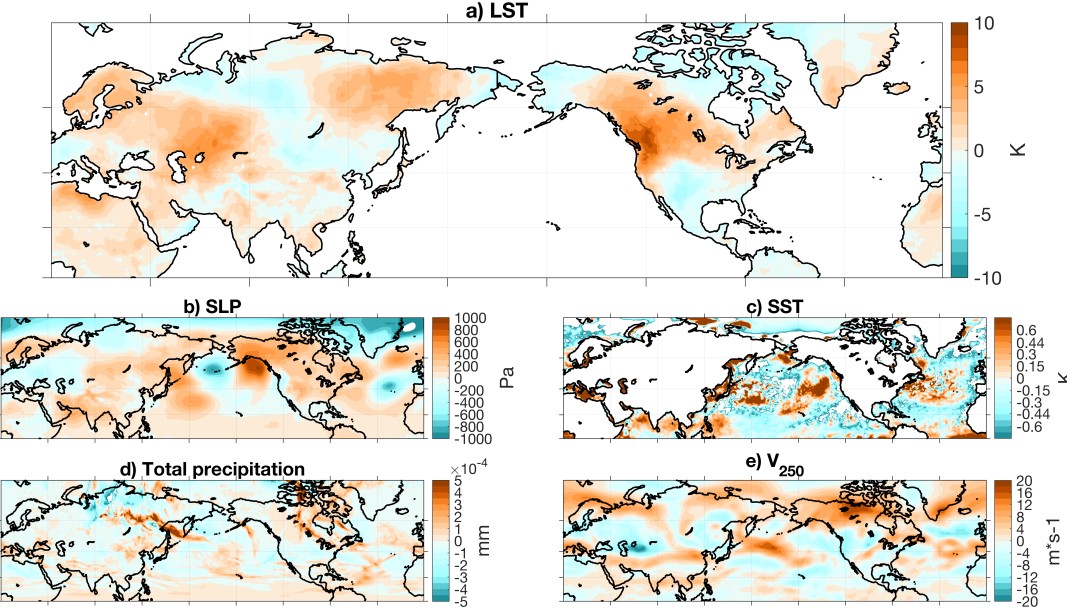

**Figure 2.** Composite mean anomaly maps for 24/6/2021-05/07/2021 of: a) LST, b) SLP, c) SST, d) total precipitation and e) wind speed at 250hPa.

in the upper mid-latitudes. These are partly compensated by strong positive contributions from $k = 5$ and $k = 8$. The role of wavenumber 3 in triggering the negative MHT anomaly, peaking at around -0.5 $PW$ is highlighted in Figure 3d. The negative anomalies emerge in the high latitudes at the beginning of the heatwave, then moving southward and reaching 50N in the next few days.

Summarizing, the 2021 heatwave over the NW Pacific coastal regions is found to be part of a hemispheric pattern of anoma-
lously warm temperatures, and was enhanced by a strong blocking over the Western Pacific, with a jet stream displaced at very high latitudes over northern Canada. The MHT was vanishing in the zonal mean, even slightly negative at the poleward edge of the midlatitude channel, and most of the negative transport was carried out by a planetary-scale $k = 3$ wave, mainly in the poleward half of the mid-latitude channel.



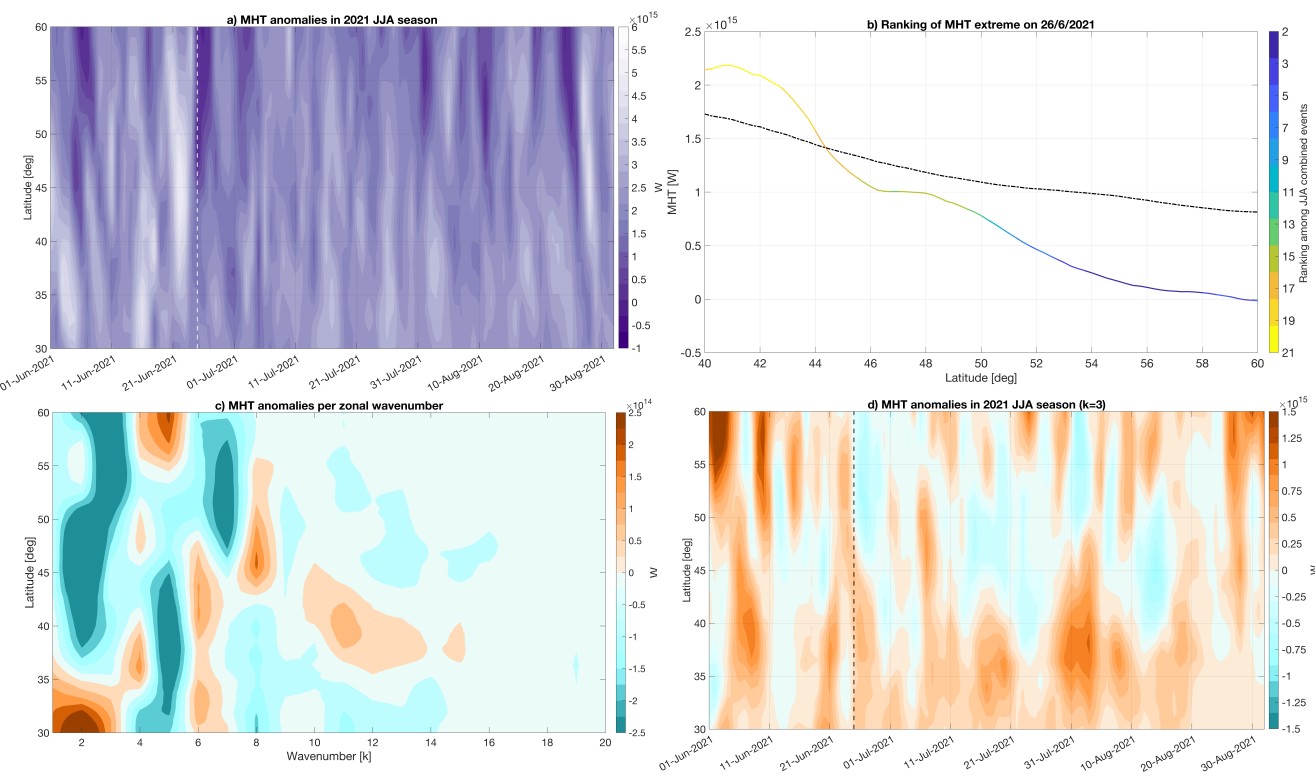

**Figure 3.** a) Hovmöller plots for total MHT during 01/06/2021–31/08/2021; b) meridional cross-section of MHT on 24/6/2021, with colors denoting the ranking position of the event in the set of all weak MHT–warm LST extremes for each latitudinal grid point; c) spectrum of MHT anomalies averaged over 24/6/2021–05/07/2021, as a function of zonal wavenumber $k$ and latitude; d) same as (a), for the anomalies in $k = 3$ contribution to MHT.



## 5 Weak MHT extremes and warm LST extremes in boreal Summer

In this section, we draw upon the case-study from Sect. 4 to investigate occurrences of $\mathrm{LST}_{95}|\mathrm{MHT}_\downarrow$ events during JJA. As indicated in Table 2, 34 days satisfy these conditions; many of these occurrences, although not all of them, span a broad latitudinal range.

The 2021 Northwest Pacific heatwave is in many aspects a typical occurrence for $\mathrm{LST}_{95}|\mathrm{MHT}_\downarrow$ events during JJA (Figure 4). Common features include:

- the largest LST anomalies are found over the western coast of North America, as well as central Asia and parts of the Arabian peninsula, although in Fig. 4 they also extend to the most part of eastern Europe;

- tropical SST anomalies are negative over much of the Pacific and the subtropical central Atlantic, as well as in the North Sea, while warm anomalies are prevalent across the western half of the northern Atlantic;

- Strong positive SLP anomalies are present over the Western coast of North America, along with western Pacific and eastern Europe;

- $V_{250}$ anomalies suggest that the jet stream is divided into two branches over western Eurasia, one spanning from the Mediterranean to eastern China and the other displaced over northern Eurasia, although patterns are not significant according to the statistical test;

As mentioned in the previous section, the pattern of SST anomalies in the Pacific is a necessary prerequisite to the formation of the low-level frontal zone advecting the diabatically heated warm air towards North America (White et al., 2023). Related to this, the SLP pattern over Eurasia favors the formation of continental blocking fueled by the jet bifurcation, which in turn enhances local Summer warming due to adiabatic compression over parts of Eastern Europe and central Asia. Together, the co-occurrence of such conditions over North America and Eurasia trigger spatially compounded heatwaves at hemispheric scale.

Comparing composite mean maps across all hemispheric LST warm extremes (Figure 9) and conditioned $\mathrm{LST}_{95}|\mathrm{MHT}_\downarrow$ (Figure 4) events, we notice that warm LST extremes tend to occur in preferred hotspots. The intensity of extremely warm LSTs and the extension of these hotspots is clearly enhanced by the occurrence of MHT weak extremes. This is facilitated by dynamical factors that are a common feature of $\mathrm{MHT}_\downarrow$ events in JJA (cfr. Figure 1b), i.e. the formation of blockings over the Pacific northwestern coast of North America and eastern Europe, amplifying local thermodynamic factors in a similar fashion as what described for the 2021 heatwave.

In order to better understand how MHT and LST extremes interact with each other, leading to the patterns described above, we look at the spectrum of MHT anomalies as a function of zonal wavenumber and latitude for all $\mathrm{LST}_{95}|\mathrm{MHT}_\downarrow$ days. Wavenumbers $k=2$ and $k=3$ exhibit a strong negative contribution (up to -0.25 $PW$) in the higher latitudes, which is partly compensated by a positive $k=5$ anomaly north of 50N (Figure 5a). The $k=2,3$ negative anomalies are sufficiently strong to result in a negative MHT for those wavenumbers, with the most extreme cases even reverting the sign of the total MHT, e.g. the 2021 described above (cf. the climatological spectrum in Figure B1).



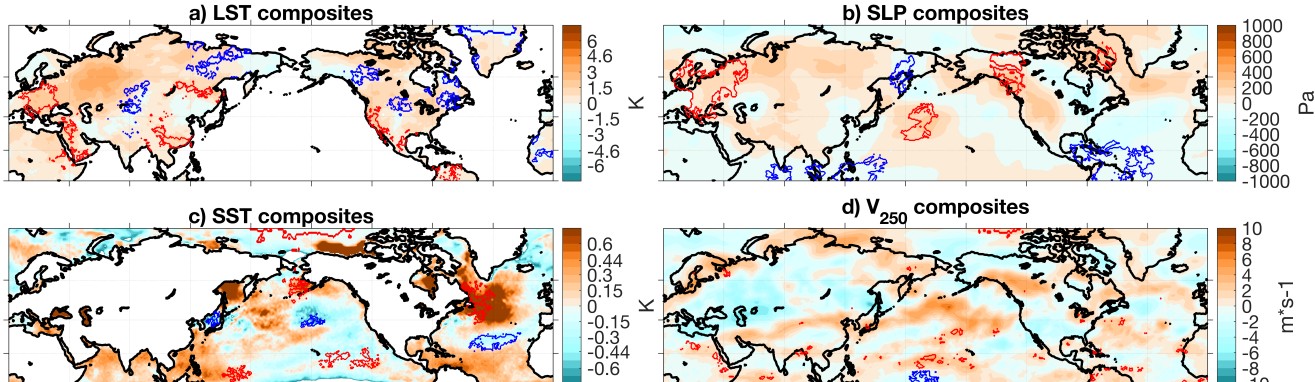

**Figure 4.** Composite mean anomaly maps of a) LST, b) SLP, c) SST and d) $V_{250}$, averaged across all days featuring an extremely weak MHT and an extremely warm hemispheric LST anomaly in JJA. The red/blue contours denote significant anomalies with respect to the statistics of all $LST_{95}$ daily events.

## 6   Weak MHT extremes and warm LST extremes in boreal Winter

We now turn our attention to $LST_{95}|MHT_\downarrow$ events in the boreal Winter (DJF). Figure 6, analogously to Figure 4, displays composite mean anomaly maps of LST, SLP, SST and $V_{250}$ for these conditioned events in DJF. We identify several salient features:

- The warmest LST anomalies are centered in continental areas of Siberia and Canada;

- SLP anomalies display extended low pressure systems over the continents, coherently with an AR/PNA- pattern (cfr Lembo et al. (2022));

- Cold SST anomalies are present over the eastern Pacific and central Atlantic;

- An anomalously strong jet stream is found over the central Pacific and western Europe;

The tropical Pacific displays strong SST warm anomalies. Together with the anomaly in the jet stream, this hints at the occurrence of a westerly QBO phase (cfr. Kumar et al. (2022) for related implications regarding the subtropical jet stream) related to a positive ENSO3.4 index.

The distribution of MHT across the spectrum of different zonal wavenumbers (Figure 5b) highlights a dominant anomalously negative $k = 2$ contribution in the middle and high latitudes (up to more than -0.5 $PW$). Unlike in JJA, this is neither sufficient to revert the sign of the transport in that specific wavenumber, nor to determine a total negative MHT (cfr. Figure B2). Nevertheless, the strongest $k = 2$ extremes are sufficiently large to result in an almost vanishing total MHT. This is coherent with the patterns highlighted by the composite maps (Figure 6b,d); the extended low pressure systems at high latitudes over the continents result in an anomalously strong subpolar jet. The phase of the $k = 2$ wave reflects this, favoring the formation





**Figure 5.** Spectrum of MHT anomalies ($W$) averaged in JJA (a) and DJF (b) across all days featuring a $\mathrm{LST}_{95}|\mathrm{MHT}_\downarrow$ event, as a function of wavenumber $k$ and latitude.

of strong ridges over the Pacific ocean, North Atlantic and the Mediterranean. The jet stream is strengthened and displaced at the northern boundary of these blockings, fueling an enhanced zonal regime (cfr. Figure 1e).

     In order to facilitate a process-based understanding of $\mathrm{LST}_{95}|\mathrm{MHT}_\downarrow$ occurrences, we select a remarkable case-study in DJF, similarly to what was done for JJA. More specifically, we focus on the event that occurred on 19th January 2007 (Figure



8a), associated with a 2-weeks long hemispheric heatwave occurred between 19th January 2007 and 1st February 2007. Such
heatwave contributed to the largest DJF mean warm anomaly ever recorded in central Eurasia until 2020, and was characterized
by abnormally low SLP values along the Eurasian Arctic coast (Koenigk and Fuentes-Franco, 2019). The extremely weak MHT
anomaly, that ranked first among $LST_{95}|MHT_{\downarrow}$ events between 40N and 50N (Figure 8b), resulted in an almost complete
suppression of the poleward heat transport, whose footprint can be found in the distribution of MHT averaged 5 days after
the event across zonal wavenumbers (Figure 8c). The combined effect of $k = 2$ and $k = 3$ determined an abnormal negative
anomaly that was only partly compensated by the positive anomaly associated to $k = 7$. Both $k = 2$ (Figure 8d) and $k = 3$
quickly recover from the negative anomaly, with negative anomalous transports extending from the middle of the channel
towards the high latitudes and then vanishing.

Composite maps (Figure 7) evidence that largest LST anomalies were found over central-eastern Siberia and central Asia (in
agreement with Koenigk and Fuentes-Franco (2019)), although unusual seasonal warmth was also present in Alaska and parts
of Arctic Canada. The lowest SLP anomalies were located over Western Russia as well as in the central Pacific, while a ridge
spans over the Eastern Atlantic between Greenland and Europe. Consistently, strong positive $V_{250}$ anomalies are centered over
the North Sea and extend towards Eastern Europe and across most of the subtropical regions from Japan to the central Atlantic,
suggesting an enhanced jet splitting. Cold SST anomalies are found over the Eastern Pacific, while warm anomalies are present
over the Eastern North Atlantic and off the US east coast. These anomalies are consistent with a weak positive ENS0 phase
and a weakening westerly QBO phase.

Although going beyond the scope, we briefly highlight that the winter and spring anomalies in 2007 have been linked
to the exceptional September 2007 sea-ice minimum. Particularly, Graversen et al. (2011) highlighted the role of heat and
moisture convergence towards the East Siberian Arctic and Laptev Sea, inducing a net deficit in sea-ice that had inter-seasonal
implications through the sea-ice-albedo feedback in spring, when the sea-ice extent reaches its peak. While Zhang et al.
(2008) evidenced the role of all-sky preconditioning in early 2007 for the later sea-ice melting, Sedlar and Devasthale (2012)
and Graversen et al. (2011) emphasized the role of downwelling longwave radiation, related to surface temperature warm
anomalies, substantially limiting the sea-ice growth. The suppression of MHT throughout the mid-latitudes highlighted in
this boreal Winter case study is coincident with the initiation of strong positive energy convergence to the Arctic, which is
consistent with the discussed role of atmospheric MHT for the Arctic amplification Graversen et al. (2008), particularly by
means of planetary-scale waves, as highlighted in Graversen and Burtu (2016).

## 7 Discussion

Despite the differences between Summer and Winter boreal seasons in the wavenumber-decomposed MHTs and the associated
atmospheric circulation (cf. Lembo et al. (2019, 2022)), the scales of the transport related to spatially compounding extremely
warm events are similar ($k = 3$ and $k = 2$, respectively). These involve a sometimes negative (i.e. equatorward) transport,
counteracting an overall positive transport at other scales. However, since the wavenumber-decomposed transports only provide
hemispherically-aggregated information, we are not able to identify the regions where the transport is negative or positive for





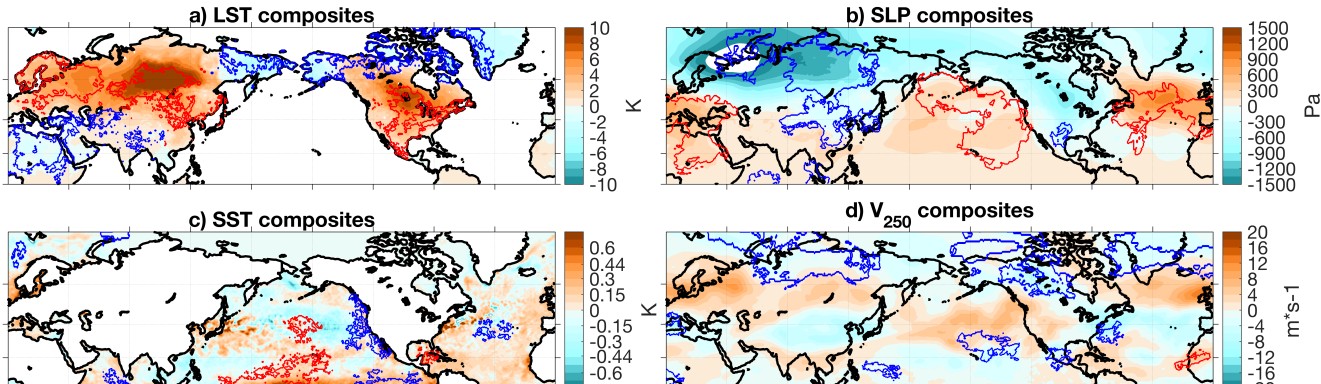

**Figure 6.** Composite mean anomaly maps of a) LST, b) SLP, c) SST and d) $V_{250}$, averaged across all days featuring an extremely weak MHT and an extremely warm hemispheric LST anomaly in DJF. The red/blue contours denote significant anomalies with respect to the statistics of all LST$_{95}$ daily events.

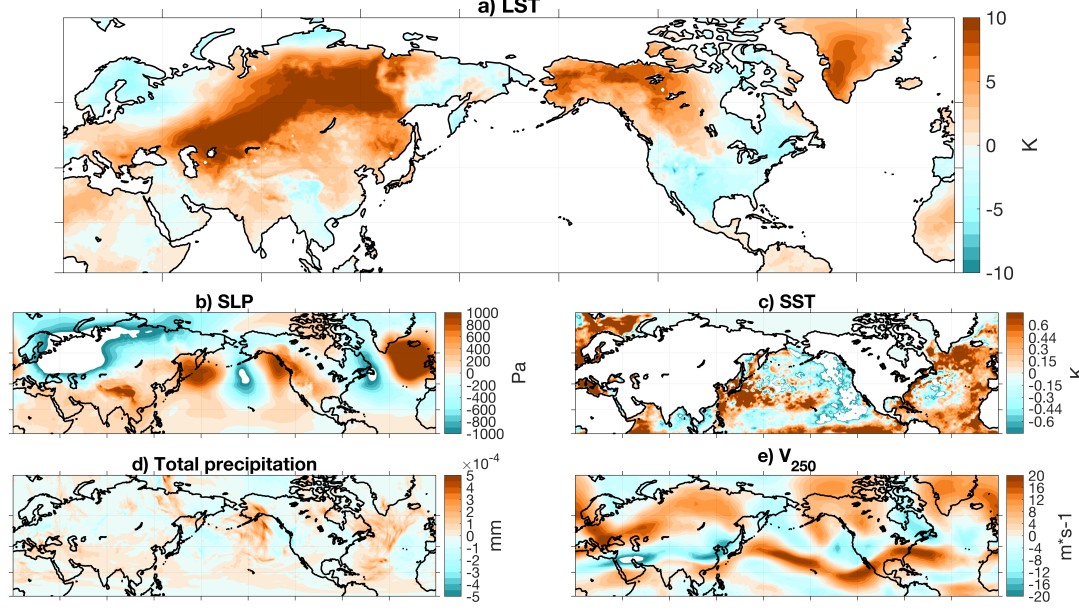

**Figure 7.** Same as in Figure 2, for the 19/1/2007-1/2/2007 period.

specific wavenumbers. Therefore, we show composite mean maps of LST$_{95}$ unconditioned events in Figure 9 and compare them with those in Figures 6-4 to address the role of weak MHT extremes for the occurrence of hemispheric heatwaves.



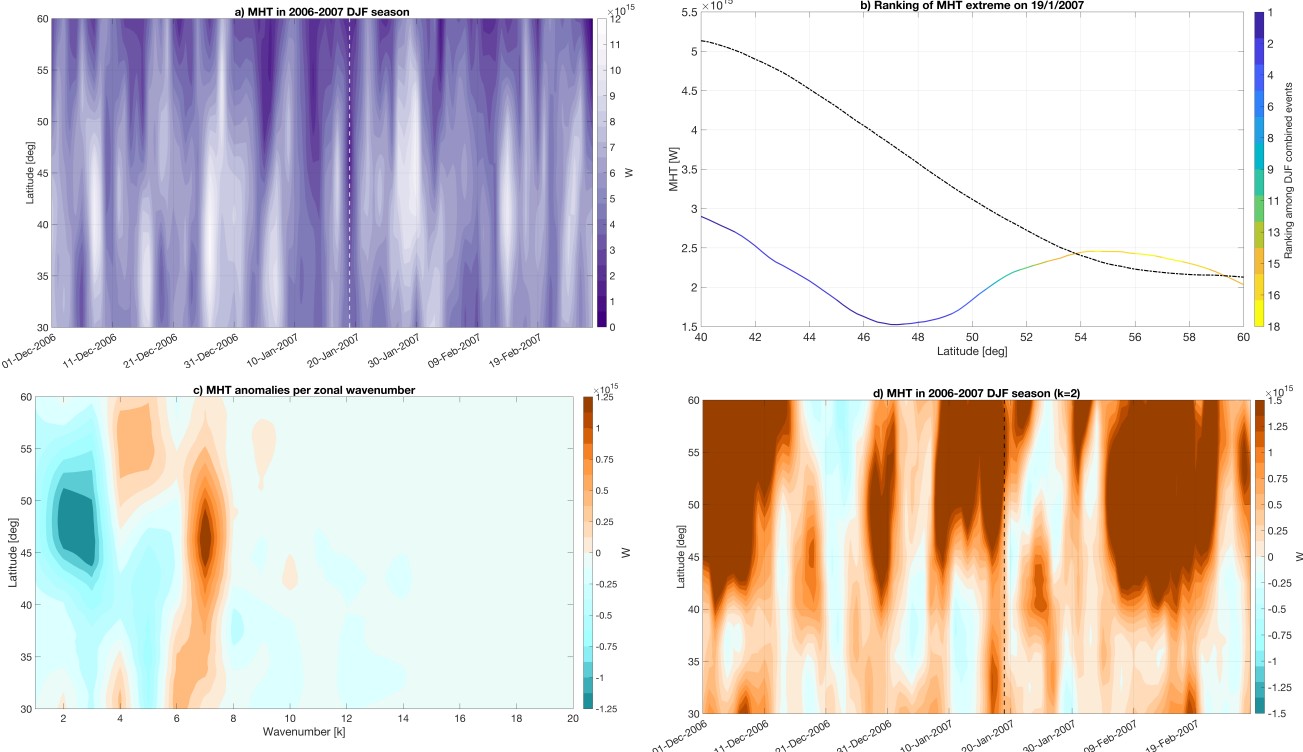

**Figure 8.** a) Hovmöller plots for total MHT during 01/12/2006–28/02/2007; b) meridional cross-section of MHT on 19/1/2007, with colors denoting the ranking position of the event in the set of all weak MHT–warm LST extremes for each latitudinal grid point; c) spectrum of MHT anomalies averaged over 19/1/2007–23/1/2007, as a function of zonal wavenumber $k$ and latitude; d) same as (a), for the anomalies in $k = 3$ contribution to MHT.

In boreal Summer, the dipolar structure of warm/cold SST anomalies over the west/east Pacific, and positive SST anomalies over North Atlantic, appear as defining features of warm $\text{LST}_{95}$ events. The high SLP anomalies over eastern Pacific and the Rockies, stretching along North Atlantic and Europe, are instead shared with $\text{MHT}_{\downarrow}$ events (cf. Figure 1). The jet stream bifurcation is a defining feature of most hemispheric heatwaves. The subtropical branch of the jet extending across Eurasia into the western Pacific, supports the formation of the frontal zone in the middle of the Pacific, which was a key feature of the 2021 Northwestern Pacific heatwave. The LST anomalies bear remarkable analogies but also significant differences between conditioned and unconditioned hemispheric heatwave events (cf. Figures 4a and 9a). Ridges over western Russia and western Canada in fact determine an increase towards the western edges of main continents of the areas affected by extreme heat, most prominently towards Eastern Europe and the Pacific North American coast.

Such location of ridges and troughs in the upper half of the mid-latitudes, and in particular the significance of ridges in the $\text{LST}_{95}|\text{MHT}_{\downarrow}$ composite mean patterns, denote a peculiar phase of the $k = 3$ planetary wave, favoring the emergence of hemispheric heatwaves. In fact, ridges placed over continents, unlike over oceans, favor unusual warming in the high latitudinal



continents and prompt advection of heat southwards towards mid-latitudes on their eastern edge. This is the case especially in Summer, when continental regions are prone to localized warming due to subsidence, most notably in dry environments (Black et al., 2004; Bieli et al., 2015). Changes in frequencies of dominant weather regimes (Lembo et al., 2022) support this argument, with ridges over the Western North America, and co-occurring Scandinavian and Siberian blockings being recognized as a defining feature of $\mathrm{MHT}_\downarrow$ events in JJA.

In Winter, different behaviors are identified over the Pacific and Eurasian-Atlantic sectors, in a context of generalized enhanced zonality that is found for both conditioned (Figure 6b and unconditioned events (Figure 9f). Regarding the Pacific, the ridge emerging in conditioned events is replaced by a dipolar structure, with positive anomalies over the western Pacific, negative over the eastern Pacific. The conditioned pattern is in fact rather a result of the land-sea phasing of the $k = 2$ wave in the Pacific region mentioned above (cf. Section 6). Unlike unconditioned events, this leads to relatively cold SST anomalies along the western North American coast, that at the same time prompt positive LST anomalies across much of central North America. Regarding the Eurasian-Atlantic sector, the Atlantic ridge extends further eastward over the Mediterranean in the conditioned composites, and the deep low SLP anomaly centered over the Kara Sea stretches along much of Northern Eurasia, accompanied by widespread positive $V_{250}$ anomalies. In other words, while the excessive zonality over Eurasia is a common feature of unconditioned $\mathrm{MHT}_\downarrow$ (Figure 1f; also compare dominant NAO+ and AO weather regimes in Lembo et al. (2022)) and $\mathrm{LST}_{95}$ events, the vanishing $k = 2$ contribution related to $\mathrm{LST}_{95}|\mathrm{MHT}_\downarrow$ events results in an extensive warming of eastern Europe, eastern Siberia and inner Mongolia, that would be less relevant otherwise (Figure 9e). Summarizing, the combined occurrence of extremely weak MHTs and hemispheric LST warm extremes enforces a pattern of hemispheric-wide excessive zonal circulation, associated with a vanishing heat exchange between low and high latitudes. Temperature anomalies suggest that zonal gradients between land and oceans are damped by this zonal circulation, emphasizing meridional temperature gradients between the Poles and mid-latitudes.

Assuming that in the next future the Arctic amplification will act by reducing the meridional temperature gradient in a permanent way, we could hypothesize that this will result in a gradual shift of the $k = 2$ role for MHT extremes from a DJF-like situation, in which enhanced zonal circulation will bring mild air towards the continents, towards a JJA-like situation, with advection of diabatically heated air from the oceans towards the western boundaries of the continents mediated by $k = 3$ suppression or reversal, especially over the Pacific. In this respect, a focused examination of the tropical intrusions and atmospheric rivers toward the western sides of the continents will be crucial in order to disentangle the thermodynamic and dynamic contributions to the MHT extremes, and highlight the role of diabatic flow for nonlinear Rossby waves amplification (cfr. Grams and Archambault (2016)). On the other hand, as described regarding the winter 2007 event, we do expect that the partition of energy transport between mid-latitudes and the Arctic will be modified by these mean state changes, especially regarding the contribution of planetary scales and latent energy.

Finally, our results support previous findings on the typicality in the development of concurrent heatwaves, as highlighted for DJF European heatwaves by Galfi and Messori (2023) and for the 2021 NW Pacific heatwave by Lucarini et al. (2023). That is, that record-breaking events need not result from unprecedented physical drivers, and they can be "typical" with respect



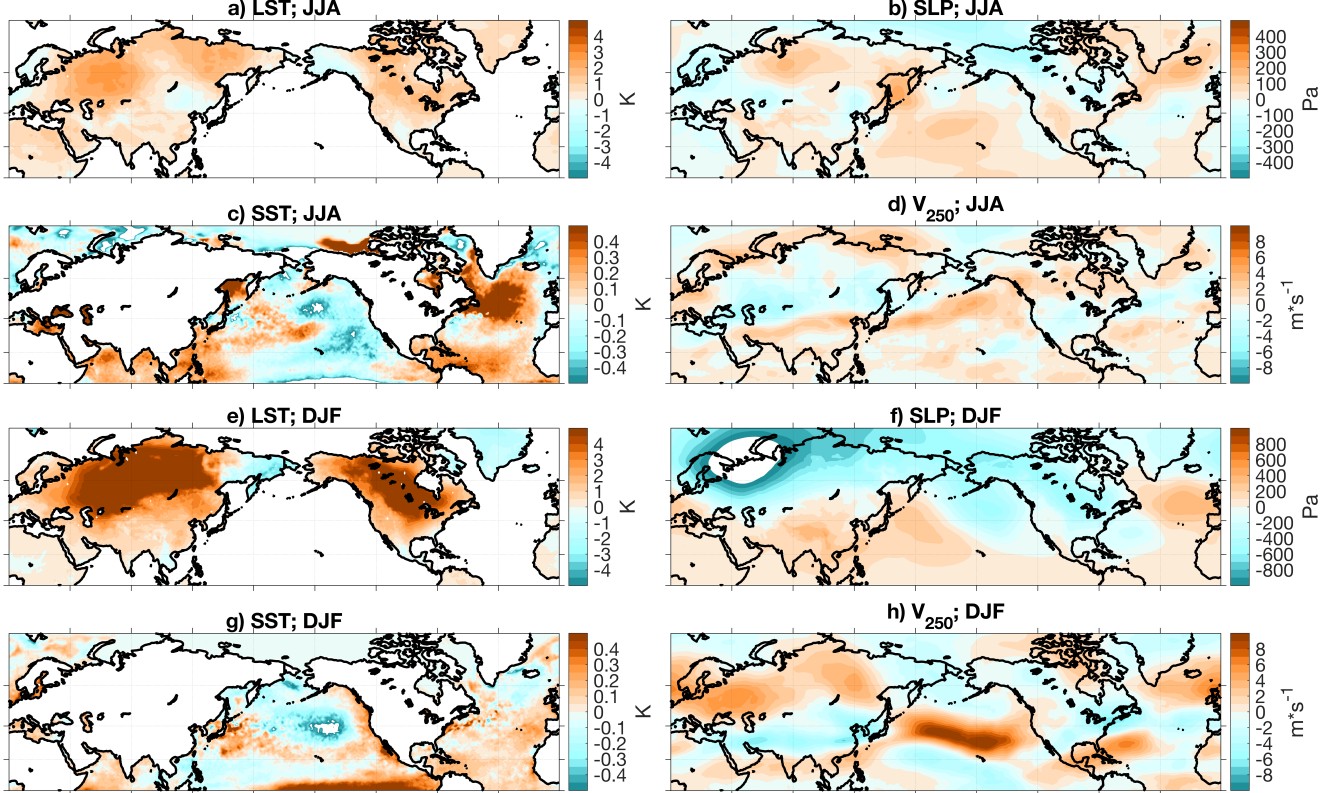

**Figure 9.** Composite mean anomaly maps of LST (a,e), SLP (b,f), SST (c,g) and $V_{250}$ (d,h) anomalies averaged over all $LST_{95}$ events in JJA (a-d) and DJF (e-h).

to other extremes of comparable intensity. We extend here these arguments about typical concurrent heatwaves to the dominant patterns of the hemispheric circulation.

## 8  Conclusions

We investigated the relationship between co-occurring heatwaves in the Northern Hemisphere mid-latitudes and meridional heat transport (MHT). We found that the conditional occurrence of extremely weak MHT and extremely warm hemispheric
land surface temperature (LST) was much more frequent than the other combinations in both the winter and summer seasons, and accordingly we focused our analysis on these cases.

During Summer extremely weak MHT – warm LST events are associated with equatorward (negative) MHT from wavenumber 3. This is sufficiently strong to cause a negative or near-zero overall hemispheric MHT. The anomalous wave-3 transport coincides with atmospheric blocking over the Northwestern Pacific region of North America and western Russia, leading to
extremely warm anomalies over the Pacific coast of North America, eastern Europe and parts of Siberia. The record-breaking



2021 Northwestern Pacific heatwave was a typical occurrence of such hemispheric circulation, albeit record-breaking in intensity, and was associated with extremely weak MHT.

In Winter, extremely weak MHT and warm LST events result from the wavenumber 2 contribution to the overall almost vanishing MHT. The anomalously weak MHT carried by this wave is related to its phase, denoted by the presence of ridges over
the ocean basins and zonally extended troughs over the continents. We argue that the zonality of the atmospheric circulation leads to the suppression of MHT and the unusual warming of continental regions through the advection of moist and mild oceanic air.

We highlight that the conditional occurrence of extremely weak MHT and warm hemispheric LSTs is relatively frequent. Indeed, 42.3% and 30.4% of extremely warm hemispheric LST days coincide with a weak MHT extreme, for DJF and JJA,
respectively.

This work has shown that co-occurring heatwaves at hemispheric scale and the general circulation of the atmosphere are to a great extent related to each other by means of the strength of the heat exchanges between the Poles and the Tropics, carried out in the mid-latitudes through eddies. Planetary-scale eddies are in this respect crucial, and we highlighted that wavenumbers 2 and 3 and their modulation are instrumental in shaping the simultaneous occurrence of circulation patterns
that favor the development of heatwaves at the regional scale. These findings suggest that a careful evaluation of anomalies in such planetary-scale waves may further improve our capability to understand and predict concurrent heatwaves, both in boreal Summer and Winter.

## Appendix A: Statistical tests for the significance of composite mean anomalies

In principle, a statistical test could be performed at every grid point of a two dimensional field, applying a correction for the
false discovery rate (Wilks, 2016). However, in cases involving a large number of simultaneous tests (approximately 130,000 in this study), such corrections result in a substantial loss of statistical power, i.e., the ability of the test to reject the null hypothesis when it is false. Since our primary goal is to identify regions of significant change without requiring fine spatial resolution, we aggregate grid boxes into clusters and perform a single statistical test for each cluster. This approach improves statistical power at the cost of spatial detail.
The clustering methodology can be summarized as follows:

1. grid points for which at least 66% of the time steps within the LST|MHT events sample exceed or fall below the time-averaged observable for the corresponding grid point are selected;

2. a density-based methodology (Ester et al., 1996) is adopted to cluster neighboring grid points;

3. clusters that are within a radius of $\epsilon$ grid points from each other are grouped and considered as a single cluster;

4. only clusters exceeding a given size $g_{min}$ (in grid points) are selected;

5. a spatial average is performed over each selected cluster or cluster group to obtain time series;





**Table A1.** Parameters for the density-based clustering algorithm.

|         | $g_{min}$ | $\epsilon$ |
|---------|-----------|------------|
| LST     | 100       | 20         |
| SST     | 100       | 10         |
| SLP     | 200       | 10         |
| $V_{250}$ | 150     | 10         |

Parameters $\epsilon$ and $g_{min}$ for each observable are given in Table A1. The choice of the parameters is motivated by the different underlying dynamics that we aim to evidence. For instance, surface temperatures are more prone to be affected by complex topography, orography and coastlines than $V_{250}$, hence the smaller $g_{min}$ and larger $\epsilon$ to preserve the spatial structure.

Mean values of the time series derived from the LST|MHT and LST|$\overline{\text{MHT}}$ samples are compared using a two-sided Student's t test at a $\alpha = 0.05$ significance level, applying Welch's modification to account for unequal variances in the two samples. The null hypothesis is that the composite mean of the field in the LST|MHT sample is equal to that of LST|$\overline{\text{MHT}}$, while the alternative hypothesis posits that the conditional occurrence of extreme MHT and hemispheric LST events significantly alters the value of the field. The case of $V_{250}$ requires a different procedure: to assess changes in the jet stream, we consider possible

differences in the sample 95th percentile of $V_{250}$, which cannot be compared using a Student's t test. To address this, we rely on a non-parametric bootstrap test instead. For each cluster, $N = 5000$ random samples of size $n$ ($n$ being equal to the number of time steps in the LST|MHT sample) are drawn from the full time series. For each bootstrap sample, the empirical 75th percentile is computed. Then, the 75th percentile of $V_{250}$ in the LST|MHT sample is compared to the distribution of the $N$ bootstrap replicates of the same quantity under the null hypothesis. If the LST|MHT value falls in either of the $\alpha/2$ or $1 - \alpha/2$

tails of the distribution, we reject the null hypothesis, implying that MHT have a significant effect on $V_{250}$ within the cluster.



Since the MHT extremes were selected using declustering (cfr. Lembo et al. (2022) and Section 2.2.2), no pre-whitening or other corrections are applied to address serial correlation. However, since the significance test is repeated for each cluster, and the number of clusters can be of the order of a few dozens, the Benjamini-Hochberg (BH) correction (Benjamini and Hochberg, 1995) with a $0.1$ significance level is adopted to control the false discovery rate. This correction is appropriate under
the assumption of independent or positively correlated tests, a condition generally satisfied by atmospheric data (Wilks, 2016).

Although a method that allows for generally dependent tests is available, known as Benjamini-Yekutieli correction (Benjamini and Yekutieli, 2001), the latter has been shown to drastically reduce the power of the test in the case of spatially extended climatological data, for which the BH version performs best (Cortés et al., 2020).

## Appendix B: Composite mean anomalies related to weak MHT extremes in the 40N-50N latitudinal band

Figure B1 displays composite mean maps of weak MHT extremes located in the 40N-50N band, similarly to Figure 1 for the 50N-60N band.

In JJA, no significant anomalies emerge, although the spatial anomaly patterns broadly match those shown in Figure 1a-d. Particularly, we notice negative LST anomalies over the southern coast of Alaska, and positive SLP anomalies across the northeastern portion of the Pacific ocean.

In DJF, there is a degree of overlap between significant regions highlighted in Figure 1e-h and those shown in Figure B1e-h. Positive LST anomalies span wide parts of Eastern Europe, China and central North America. SLP anomalies differ from Figure 1f concerning the latitude of the highest SLP in oceanic ridges, with only the Canadian trough appears to significantly differ from climatological variability, while the Scandinavian lobe of the circulation having less relevance. Besides that, a trough over eastern Asia appears to be significant, while it is not present in Figure 1f.



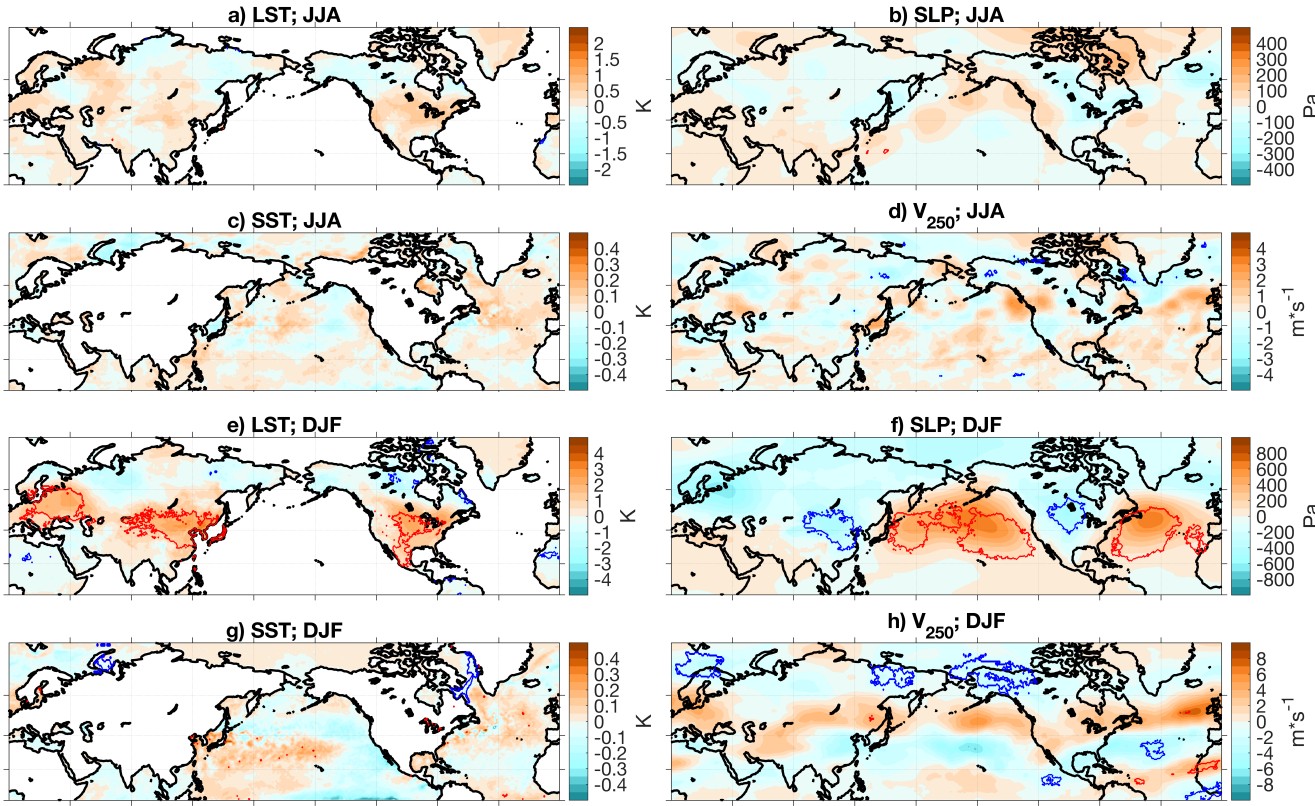

**Figure B1.** Same as in Figure 1, for weak MHT extremes located in the 40N-50N latitudinal band.



## Appendix C: Climatological spectrum of MHT

The climatological spectra of MHT as a function of latitude and zonal wavenumbers are shown for JJA and DJF in Figures C1a and C1b, respectively. Both seasons display their maxima for $k = 2$, located for JJA at the equatorward edge of the channel, and for DJF around 55N. Higher wavenumbers differ qualitatively and quantitatively in the two seasons, with JJA exhibiting weaker relative power compared to the $k = 2$ contribution than DJF. The maxima are progressively displaced from higher towards lower latitudes going from larger towards smaller wavenumbers, with most of the MHT intensity concentrated within the $k = 3 - 11$ wavenumbers range. Dominant wavenumbers, climatologically, are $k = 6$ and $k = 7$ for JJA, and $k = 3$ and $k = 6$ for DJF. This is in agreement with previous analyses of the main features of MHT in the two seasons (Lembo et al., 2022).

*Data availability.* ERA5 (Hersbach et al., 2020) was downloaded from the Copernicus Climate Change Service (2023). The results contain modified Copernicus Climate Change Service information 2020. Neither the European Commission nor ECMWF is responsible for any use that may be made of the Copernicus information or data it contains.

*Author contributions.* VL, GM and DF designed the analysis, VL carried out the main coding work and wrote most of the manuscript, VMG performed the extreme value detection and clustering, RGG performed the spectral decomposition, FEP designed the statistical significance tests and supervised the statistical analysis.

*Competing interests.* The authors declare that they have no conflict of interest.

*Acknowledgements.* VL received funding from the Italian Ministry of Education, University and Research (MIUR) through the JPI Oceans and JPI Climate "Next Generation Climate Science in Europe for Oceans" – ROADMAP project (D.M. 593/2016) and from the European Union's Horizon Europe research and innovation program Grant No. 101081193 (OptimESM). GM acknowledges support from the European Union's H2020 research and innovation programme under European Research Council grant no. 948309 and from the Swedish Research Council Vetenskapsrådet under grant no. 2022-06599. Davide Faranda acknowledges the support of the COST Action FutureMed CA22162 supported by COST (European Cooperation in Science and Technology), an INSU-CNRS-LEFE-MANU grant (project CROIRE), state aid managed by the National Research Agency under France 2030 bearing the reference ANR-22-EXTR-0005 (TRACCS-PC4-EXTENDING project), and the European Union's Horizon 2020 research and innovation programme under grant agreement no. 101003469 (XAIDA). High-performance computing and storage resources were provided by Sigma2 – the National Infrastructure for High Performance Computing and Data Storage in Norway through projects NS9063k and NN9348k.



**Figure C1.** Spectrum of MHT as a function of zonal wavenumber and latitude, in a) JJA, b) DJF.

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
