# Peer review of "Understanding concurrent heatwaves from a meridional heat transport perspective"

_EGUsphere, 2025_

## Author Comment (AC2)

**I enjoyed your paper and your methods were quite interesting. I just have a few suggestions for making it more readable:**

*We thank you very much for the constructive comments. We plan to take them fully into account in a revised version of this manuscript. We answer briefly to the specific points raised in the following:*

1. **Elaborate on the wavenumber decomposition section. In particular, I was confused by your discussion of the k=0 correction. It was difficult to find an explanation even after tracing through your citations.**

*Thank you for the comment. Indeed, this point was raised by Reviewer 1 and calls for additional effort to clarify the methodology, that was briefly mentioned with a reference to previous work. We plan to revisit the manuscript accordingly.*

2. **Elaborate on the selection of meridional heat transport (MHT) extremes. In particular, the processes of declustering using an "extremal index" and determining thresholds by latitude band using "[An] EVT_based convergence algorithm."**

*Thank you again for the comment. This is related to my previous reply. The methodology was extensively described in previous work but a revised manuscript needs to take these aspects into account in a more detailed way.*

3. **Elaborate on how you calculated the anomalous MHT. It would be nice to have a short description on your process (e.g. whether you de-seasonalized) and a statement spelling out how a negative anomalous MHT translates to a weak (less positive or slightly negative) actual MHT.**

*Same as above. We will revise the manuscript in order to clarify these aspects of the methodology in a more extensive way.*

---

## Author Comment (AC4)

*We thank the reviewer for their critical appraisal of our work. We do believe that they call on for a much more thorough effort on our part in order to justify the motivation of the work and the contribution it provides to the scientific advancements in the field of hemispheric heatwaves and their underlying dynamics. This point was also partially highlighted by the second reviewer.*

*In order to improve the clarity of our manuscript we will substantially restructure it in a more organized way by:*

*1) reducing the material presented so that the statistical connections clearly emerge from the analysis;*

*2) Supporting the statistics with physical understanding of the statistics presented, introducing a focused analysis of blockings and highlighting the climatology of absolute fields to compare with the composite anomalies thereby shown;*

*3) clearly highlighting the limitations and relevance of our study, remarking that we address occurrences of concurrent heatwaves for which the role of meridional heat transports extremes (that had never been addressed before in literature) can be justified physically with the help of specific case studies.*

*To this aim, we propose a revised structure that will read like this:*

- *An Introduction, explicitly mentioning in a clear and revised way what are the aims and scope of the present work, and what is its novelty compared to existing literature;*
- *Data and Methods;*
- *Results: for conditional occurrences, separately for boreal Summer and boreal Winter;*
- *A discussion section providing a physical interpretation of results, making explicit connections with dominant planetary scale waves, building upon previous work, with the help of case studies that will be detailed in the Supplementary material;*
- *Summary and Conclusions: evidencing the main take-home message of this manuscript, and the lines of research that are opened by this analysis;*
- *Figures 1, 4, 5, 6, 9, while those regarding case studies will be moved to the supplementary material;*

*We answer briefly the specific points raised by the Reviewer, and plan a restructuring of the way in which we present our results to support clarity on these important points:*

**At the same time, it remained unclear to me what I am supposed to learn. In other words: what is the science question that you address?**

*We wish to understand whether and how concurrent heatwaves are linked to extremes in meridional heat transport (MHT). This provides an important and timely thermodynamic complement to the many analyses in the literature focussing on the dynamics of concurrent heatwaves. We interpret our results in light of the established links between concurrent heatwaves and features such as large-amplitude planetary waves and blocks.*

**I found bold causal inferences in the interpretation of the results, which in my eyes cannot be made based on a purely statistical analysis. When stripping away these causal claims, the reader is left with a number of statistical associations, but (as I said before) I was left wondering why these are interesting or important and in what sense they contribute to an understanding.**

*In the conclusions we highlight statistical associations and conditional anomalies. We interpret these findings using terms such as "related to each other" and " These findings suggest". We have purposefully avoided any statements that suggest causal analyses that we have not performed. While we do write that "planetary-scale eddies [..] are instrumental in shaping the simultaneous occurrence of circulation patterns that favor the development of heatwaves at the regional scale", we believe this statement is supported by the analysis of the circulation anomalies in Figs. 4 and 6, which reflect established physical mechanisms leading to heightened temperatures at the surface, which we expound in terms of temperature advection from ridge and trough patterns and connections to large-scale SST anomalies.*

*We will review carefully the results and discussion sections to ensure that the text is fully aligned with that of our conclusions, and if we do chose to make causal claims, we will provide a clear reference to the parts of our analyses that support them.*

**That's too bad, because the title says that enhanced "understanding" is the main point of the paper. My perceived lack of new insight weighs even more strongly given my feeling that the key question (see my first paragraph) remains unaddressed and unanswered. It may well be that this manuscript contains important insight, but it was impossible for me to recognize it from the current presentation. My central question is: what does this manuscript contribute to our understanding of hemisphere-scale land surface heat anomalies?**

*We provide below some key conclusions of our analysis. We highlight that these are new findings, which have not previously been presented in the peer-reviewed literature.*

- *Extremely weak (in some cases equatorward) MHT can co-occur with anomalously warm land surface temperatures during both boreal summer and winter.*
- *Weak MHT events account for a considerable fraction of extremely warm land surface temperature events. These events typically display several concurrent regional heatwaves.*
- *These combinations of weak MHT and anomalously warm land regions can be understood in terms of anomalous patterns of meridional and zonal flow linked to planetary-scale wave patterns and blocking, and are consistent with the dynamical understanding of concurrent heatwaves proposed in the peer-reviewed literature.*

*We take full responsibility for these not emerging clearly from the current text and presentation of results, and will restructure the abstract, introduction and conclusions to make these emerge more clearly.*

**Possibly there is a hint at the underlying science question on line 48, namely that the authors "seek to understand how concurrent heatwaves…. are linked to extremes in planetary-scale Rossby wave anomalies….", but in my eyes this goal is not achieved. The authors at best describe that concurrent heatwaves are linked to extremes in the meridional heat transport, but in my judgment this does not result in a deeper "understanding how" this works. In other words, the authors fall short of the goal set by themselves.**

*We think that the Reviewer is being unjustifiably harsh in this comment. We first conduct a detailed statistical analysis to show that there is a link (the "describe" part), and we then dedicate considerable space to interpreting such a link in view of dynamical circulation features and large-scale modes of variability that have previously been linked to (concurrent) heatwaves. All the results sections and the discussion section provide a detailed contextualisation, for both a case study and the composite picture. A non-exhaustive list of examples include: ll. 177-187; ll. 219-224; ll. 246-256.*

*We could repeat existing analyses in the literature, for example diagnosing surface temperature advection or radiative anomalies at the surface, but we see little added value in this given the extensive work that has already been published on these topics. In general, we plan on restructuring the whole results section to provide a clearer connection between the thermodynamic statistical analysis and its interpretation in terms of physical drivers of surface temperature anomalies.*

*In addition, I have an issue with the part of the interpretation that concerns the occurrence of "troughs" and "ridges" on anomaly plots: a negative anomaly in pressure or geopotential does not necessarily imply a trough, it may just as well be a weaker-than-normal ridge. A similar issue is related to the occurrence and strength of jets.*

*We agree with the Reviewer. We will, as suggested also by reviewer 2, establish a clearer connection to blockings by using appropriate metrics and fields, and we will provide climatologies of absolute fields to underscore the relevance of detected anomalies with respect to the baseline pattern. We will revise our dynamical interpretation of the results as needed in view of these analyses.*

**Specific comments:**

**Line 29: How can an "episode" be related to a "frequency"?**

*We do not refer to a single episode here, rather to a set of episodes conditioning the absolute frequency of dominant weather regimes across a specified region. We will review the period to make it more clearly.*

**Line 38: I find this formulation misleading: in what sense is resonance "facilitated" by the underlying conditions? It seems to me that your "underlying conditions" are simply the forcing for the waves, which (under specific conditions) undergo resonant amplification.**

*Agreed. We will revise this sentence accordingly.*

**Line 44/45: what is the difference between "dynamical drivers" and a "process-based investigation". To my taste this section contains too much jargon, and moreover I would welcome a more precise distinction between this work and previous work. Do you mean that your work merely analyses statistical connections rather than investigating the underlying dynamics? The latter could involve, e.g., linking observed phenomena with the underlying equations.**

*Existing literature focusing on "dynamical drivers" of concurrent heatwaves is provided in the previous paragraph. These previous works have generally aimed to discuss what drives the occurrence of hemispheric heatwaves (in this sense, trying to establish a causal link between the drivers and the heatwaves). We do not aim to do so, but rather provide a statistical and physical understanding of how concurrent heatwaves are linked with meridional heat transport extremes via planetary-scale waves. In order to do so, we*

*will add to the revised manuscript a more in-depth analysis of some crucial physical processes, namely blockings, jet stream and climatology of absolute values in general to put anomalies in context. Furthermore, given that the distinction between "dynamical drivers" and "process-based investigation" may seem obscure to the reader, we will drop this sentence as part of a general restructuring of the introduction, aimed at highlighting the scope of the present work.*

**Line 63: what is a "rectilinear grid", given that your domain is spherical?**

*We meant here that the ERA5 data provided on a native Gaussian grid had been retrieved through the Copernicus Data Store service. In this case, ERA5 data are regridded on a regular lat/lon grid. We will clarify this in the revised text.*

**Line 88, "correction for the wavenumber 0": can you explain in a little more detail? What is the "zonal circulation"? You talk about a *vertical and meridional mean transport*", but isn't the key here that you are considering a *zonal mean* transport?**

*We refer to the correction for instantaneous meridional heat transports in k=0, that was introduced in Lembo et al. 2019 based on the work by Liang et al. 2018. When considering daily or sub-daily transports, the mass transport component in the enthalpy flux needs to be taken into account, as it is not negligible. We do so by removing from the k=0 component a term accounting for energy transport associated with the total mass flux, which is dependent on the zero energy reference level. We will expand this explanation in the revised text, based on this previous work.*

**Line 91: not clear to me what the word "this" refers to.**

*If the reviewer refers to line 89, we refer to the "correction" for k=0. We will make it explicit in the revised text.*

**Line 99: Does table 1 refer to MHTs including all zonal wavenumbers?**

*Indeed. We will clarify this in the revised text.*

**Line 104: At first it did not become clear to me how you perform the "spatial average" of daily anomalies. Do you take the absolute value of an anomaly before you perform the spatial average? If not, warm and cold anomalies might cancel to a large extent. I presume that you do NOT take the absolute value, and that's probably a main point of your analysis, but this should be made very clear to the reader.**

*Agreed. We obviously compute the anomalies locally before performing spatial averages. We will state it clearly in the revised text.*

**Line 111: "Impact of MHT extremes on LST extremes": this phrase is misleading, since it suggests that you can make a statement about causality…. which you cannot, based on a purely statistical analysis alone (see J. Pearl: The book of why). Correlation does not imply causation! Similarly on line 157 where you promise a "process-based understanding of mechanisms", which would imply that you can distinguish cause and effect.**

*Clearly, we are not making statements about causality, neither here, nor anywhere in the text. We will replace "impact" by "role", clarifying that we are interpreting statistical associations through the lens of what we argue are the relevant processes, given the existing literature on the topic. We do apologize once again if our intentions were not made sufficiently clear from the very beginning.*

**Line 111: What is a 2D-field at a grid point?**

*We mean here the distribution along time for every grid-point in the 2D field. We will clarify it in the revised text*

**Line 114: "composite mean of such quantities": not clear to me.**

*Apologize for the confusion. We refer here to the "composite mean" of LST, SST, SLP, V250 anomalies conditioned to the occurrence of LST|MHT events. We will clarify this in the revised text.*

**Table 2 and thereafter: is "LST | MHT" simply the number of cooccurring LST and MHT events?**

*Indeed. Absolute values and relative occurrence (in percentage) of days featuring co-occurring extremes are here shown.*

**Table 2: the last row contains bold-face entries with less than 10%, which seems to contradict the table caption.**

*Apologize. The table caption was indeed misleading. We will revise it accordingly to reflect that we also include 9.6% and 8.5% when the whole channel (30N-60N) is taken into account, to reflect that the conditional occurrence exceeds 10% in one or more 10-degrees-wide latitudinal channel(s).*

**Line140: given that according to Table 2 about 91.5% of all negative MHT extreme events in JJA are NOT associated with positive extreme LST events, it is not clear to me why the selection of events in the composites of Fig 1 is meaningful.**

*This is a fair point, that was raised by reviewer 2 as well, and we do apologize if it was not made sufficiently clear throughout the manuscript, that we do not claim that there is an absolute majority of MHT extremes that are conditioned to LST hemispheric extremes (nor the contrary). We do aim to show here (and we do so by a rigorous statistical testing), that the patterns that are shown in figure 1 are significant where highlighted by the mask. We do also argue, and we justify this by the intermittency of MHT extremes, that if the full length of each heatwaves for which at least one day featured a conditional occurrence, the number of heatwaves involved is substantial, as described in ll.130-135. We will make sure to make this point even clearer in the revised text.*

**Line 146: It seems far-fetched to associate one single local pressure anomaly with the PNA pattern.**

*We apologize for the confusion. Having introduced the conditional occurrence, it might be misleading that Figure 1 refers to the composite mean of all MHT weak extremes in the 50N-60N channel. The number of this occurrence is not negligible (around 200 events) and the statistical testing suggests that SLP anomalies in the Northwestern Pacific area are sensitive to MHT weak extremes. In line with what previously observed about weather regime occurrences (Lembo et al. 2022), we therefore claim that this significance might be related with an emerging PNA+ pattern.*

**Line 147: "with respect to JJA"?**

*Indeed. We are here comparing Figures 1b and 1f.*

**Line 148, "Accordingly": how should the features on the LST plot associate with the features on the SLP plot? For me (not having read your previous papers) this is not obvious. Later in line 150 you state that this is somewhat "counterintuitive".**

*Apologize for the confusion. Indeed, the word "Accordingly" here might be misleading and we plan to revised the sentence in order to reflect this.*

**Line 149 (and similarly in the abstract line 7), "....two deep troughs": a negative SLP anomaly is not necessarily a trough, it may just as well be a weaker-than-normal ridge.**

*Agreed. We will provide climatologies of anomalies where most suitable, either in the figures or in a supplementary material.*

**Line 151, "suppression of meridional exchange": this seems to conflict with the fact that these plots are conditioned on events with extreme (negative) MHT: can you explain to the reader who has little experience with your previous work?!**

*This sentence aims to connect SLP and V250 anomalies with the weak MHT extremes, that are by definition indicative of a suppression of meridional heat exchange. Only in very few cases, and mostly in boreal Summer, MHT extremes turn negative, as explained in Lembo et al. 2022.*

**Figure 1: I would appreciate axis labels with the respective longitude and latitude.**

*Agreed. We will provide it in the revised version of the figure.*

**Line 156: it is not clear to me how this plot indicates the location and/or the strength of the jets.**

*Apologize for the confusion. The statement about the location and strength of the jets builds upon the anomalies displayed in the composite maps in Figures 1d and 1h for V250. The significance masks for DJF suggest that negative anomalies in the subtropical Atlantic are departing from climatology, suggesting a splitting of the jet. This will be clarified, as mentioned above, by showing climatology of these fields for the whole period.*

**Line 165: what is a "statistical simulation"?**

*Apologize for the confusion. We meant here "statistical analyses (Pons et al., 2024) and numerical simulations". We will revise the text accordingly.*

**Line 165, "the above mechanisms": you just provided descriptions, I would not consider this as elucidating "mechanisms".**

*We refer here to the mechanisms described to explained the 2021 NW Pacific heatwave occurrence, not to what was described in the present work beforehand. We will state it more clearly in the revised text.*

**Line 169: the paper that you refer to does NOT associated the actual episode of the Pacific heatwaves with resonance (rather, the authors associate a *precursor* event with resonance).**

*Agreed. Apologize for the confusion. Chen et al. 2023 indeed refers to the role quasi-resonant amplification has for a precursor of the 2021 NW Pacific heatwave. We will make it clear in the revised version of the manuscript.*

**Line 178: It is not clear to me how you infer "blocking" from just looking at SLP anomaly patterns. What is your definition of "blocking"?**

*We agree that, given that we are looking at SLP patterns, it would be more correct to refer to these as high pressure or low pressure systems. In general, when looking at blocking, as we did during this analysis, but did not accommodate in the manuscript, we refer to "blocking" in the Tibaldi-Molteni classical sense, with updates from Davini et al. 2012. These indices are typically computed on 500 hPa geopotential patterns, rather than SLP. We felt that providing these explicit diagnostics of the blocking was beyond the scope of the present work, but we can realised that a revised manuscript should contain it, and we aim to do so.*

**Line 187: how do you see a "branch of the jet stream" in a plot of wind-anomalies?**

*Apologize for the confusion. We think that the anomalies are descriptive (and we reflected this in the statement "suggests a displacement of both branches of the jet stream") but as mentioned above we will corroborate this with comparison to climatology.*

**Fig 3a: Hovmoeller Plots? I can only see one plot. Also, it would be desirable to use a color scheme that clearly distinguishes positive from negative values.**

a)  *there is a typo in the figure caption, we will remove "s" in the revised text;*
b)  *A revised version of Figure 3a will be provided, highlighting the negative values, where present;*

**Line 191: what would be a typical climatological value?**

*Climatological values are those shown by latitude with the dashed line in Figure 3b. We will be more quantitative in the revised version of the manuscript.*

**Fig 3b: what is the dashed line in that panel?**

*Apologize for the confusion. See our comment above. We will make explicit what the dashed line is in the figure caption.*

**Line 192: at this point it is not clear (to me) how negative MHT can contribute to a local heat wave.**

*This is the main point we want to make as take-home message to the Discussion section, and requires collecting all the results that we are describing in this and the following sections. We will try to make it clearer through the revisited structure of the manuscript.*

**Line 196: "triggering" suggests a cause-and-effect relation, which however cannot be established from such a plot**

*The Hovmöller plot in Figure 3d displays the evolution of k3 anomalies. Comparing the magnitude of these anomalies at the starting date of the heatwave (dash vertical line) with the absolute value of MHT in Figure 3a suggests that k3 contribution is very likely the main contributor to the overall extreme. This will be made clearer by the different color coding in Figure 3a, and will provide an additional analysis displaying the evolution of each contribution in the 50N-60N during the 24/6/2021-05/07/2021 period. In any case, we agree that this does not imply a rigorous cause-effect relationship in a statistical sense, therefore we will rephrase this accordingly.*

**Discussion of Fig 3: fine, but what am I supposed to learn from it?**

*Figure 3 is instrumental to insert the case study of 2021 NW Pacific heatwave in the context of conditional hemispheric warm LST-weak MHT extremes. We complement the analysis already provided in literature with the MHT perspective, in doing so suggesting in a clearer way what the coincidence of the two extremes implies. If we would have started with analysing all cases together, given that they are different in their manifestation, although sharing the crucial traits, it would not have been possible to address this point of view in a similarly clear manner.*

**Line 200, "was enhanced by a strong blocking…": this was NOT shown**

*We respectfully disagree with the reviewer regarding this comment. We interpret the SLP anomalies in Figure 2b as the signature of a blocking. We also notice that more focus on this aspect would have been an unnecessary effort, given that the presence of the blocking was already emphasized in previous literature. Nevertheless, we plan to expand on this interpretation by showing climatology of SLPs across this region, as mentioned in previous comments. We will also provide an estimate of blocking indices, as mentioned above.*

**Line 219: it seems impossible to infer a "prerequisite" from just a single case.**

*Agree, and sorry for being misleading. We are not claiming that SST anomalies are always a prerequisite from the single 2021 case. We are referring to the literature (e.g. White et al. 2023) in which it is stated that the SST anomalies were instrumental for the development of the heatwave in the NW Pacific. This said, the main point here is taht we show that this occurrence is characteristic of conditional hemispheric warm LST- weak MHT extremes, as shown in Figure 4.*

**Line 221, "fueled by": what does this mean? A causal connection? Is the blocking caused by jet bifurcation, or is the jet bifurcation caused by the blocking, or are both caused by something else?**

*Agreed. The statement like this is relatively confusing and will be restructured with the help of climatology, blocking index and other metrics related to the jet stream. This will be part of the mentioned substantial revision of the manuscript that will be provided.*

**Line 222, 223: similar concerns ("enhance", "trigger")**

*See our previous comment.*

**Line 257: do you really select a "case study" (or maybe rather a case)?**

*Agreed. It is more appropriate to refer to it as a case, in this context. We will change it accordingly.*

**Fig 8 called before Fig 7?**

*Agreed. There has been a mismatch in the ordering of the figures on the TeX source file, leading to incorrect ordering of the referred figures in the rendered document. We will invert current Figures 7 and 8 in the revised manuscript.*

**Lines 282/283: how can a reduced MHT be associated with energy convergence into the Arctic?**

*Agreed. This point is somewhat counterintuitive, and it is not easy to draw connections between what we discussed for the mid-latitudes and what happens with the Arctic amplification. As we feel this would require additional analysis that goes beyond the scope of the present work, we will drop the sentence and restructure this paragraph accordingly.*

**Line 297, "Jetstream bifurcation" is misleading, since the figure shows anomalies. In other words, the "bifurcation" may be part of the climatology and only be slightly stronger or weaker during the considered episodes.**

*Agreed. Climatology of V250 will be provided in the revised text to address the magnitude of significant detected anomalies, as also mentioned in the previous replies.*

**Line 303 (and also abstract line 7), "location of troughs and ridges": again (like above), a map that shows anomalies cannot make a statement about the location of troughs and ridges. In other words: a weaker-than-normal ridge is not necessarily a trough.**

*Agreed. See our reply above (and others beforehand) about the climatology.*

**Line 315, "leads to": do you imply a causal connection?**

*No we do not. We agree that the phrasing might be misleading and we plan to revisit accordingly in the new version of the manuscript.*

**Line 321, "results in": ditto.**

*See our reply above.*

**Line 344: here it is not clear what is conditioned on what.**

*Agreed. The sentence will be revised in order to reflect what was actually done during the analysis.*

**Line 348, "sufficiently strong to cause….": how does this causal chain work? Do you ever in this paper give an explanation, or provide at least a hypothesis?**

*Agreed. We are aware that this was the only part of the manuscript where we actually implied that our analysis is establishing a causal link between MHT and hemispheric LST extremes. This phrasing is actually misleading and does not reflect our interpretation, therefore we are planning to change it in the revised version of the manuscript. As mentioned above, we gave an interpretation of what the mechanism linking MHT and hemispheric heatwaves is according to the present analysis, previous work and available literature. This interpretation was the main core of Section 7 and more specifically lines 303-326. We apologize for not having made this sufficiently clear and having somehow suggested that we were attempting a causal inference between MHT and LST. We take full responsibility for it and plan to revise the manuscript in order to better clarify our intentions and convey the relevant message in the conclusions section.*

---

## Author Comment (AC5)

This study explores the link between concurrent hemispheric heatwaves (high Land Surface Temperatures, LST) and extremely weak Meridional Heat Transport (MHT) events. Authors present analysis for these combinations during boreal summer and winter, describing in detail the associated dynamical patterns across multiple variables, which they relate to existing literature. This descriptive approach is thorough, but the interpretation could be strengthened by drawing clearer connections to the study's main research questions and hypotheses, which are not clearly formulated to begin with.

While the study tackles an important and timely topic, its current presentation limits the clarity and potential impact of its findings. The manuscript would benefit greatly from a clearer framing of the study's goals and novelty early on, as well as a more integrated structure that better guides the reader through the analysis. In particular, introducing the central mechanism earlier and using schematic summaries would help contextualise detailed descriptions and improve overall readability. I therefore recommend a substantial restructuring to enhance narrative flow and ensure the scientific significance of the work is fully conveyed. See general comments for suggestions intended to be constructive.

***We thank the reviewer for the insightful evaluation of the manuscript and constructive criticism. We take full responsibility for not having conveyed the main message of the analysis in a sufficiently clear way, and for the lack of structure that the reviewer perceived in the way arguments were exposed. This is in line with the criticism raised by reviewer 1, and prompt us to substantially rethink the organization of the work, and to highlight the main message in a more logical way. As mentioned in the reply to reviewer 1, the restructuring of the manuscript would lead to a tentative revised structure that would look like this:***

- ***An Introduction, explicitly mentioning in a clear and revised way what are the aims and scope of the present work, and what is its novelty compared to existing literature;***
- ***Data and Methods;***
- ***Results: for conditional occurrences, separately for boreal Summer and boreal Winter;***
- ***A discussion section providing a physical interpretation of results, making explicit connections with dominant planetary scale waves, building upon previous work, with the help of case studies that will be detailed in the Supplementary material;***

- *Summary and Conclusions: evidencing the main take-home message of this manuscript, and the lines of research that are opened by this analysis;*
- *Figures 1, 4, 5, 6, 9, while those regarding case studies will be moved to the supplementary material;*

*We answer briefly the specific points raised by the Reviewer, highlighting how we are planning to achieve this revision.*

**General Comments**

1. The introduction currently reads as though there is some confusion or inconsistency in how the authors position their work relative to existing literature and understanding. Authors write that they intend to explore link of LSTs to extremes in planetary-scale Rossby wave anomalies, which is a classic subject in atmospheric dynamics, yet at the l. 44 write "the literature [...] primarily focused on dynamical drivers," implying that in this study they will not do so, but isn't this study also examining dynamics? Perhaps the intention is to highlight that prior work has largely focused on Rossby waves as responses to atmospheric variability, without explicitly linking them to MHT processes. If so, I suggest clarifying this distinction. I recommend placing greater emphasis on the novelty of incorporating one large-scale thermodynamical mechanism (i.e. MHT). To strengthen this point, it would be helpful to include a statement highlighting that, to their knowledge, no prior study has specifically explored this linkage. That said, the study also engages with wave dynamics (e.g. wavenumber analysis), so care should be taken not to present the work as focused solely on thermodynamics. Emphasising the integration of thermodynamical and dynamical perspectives would more accurately reflect the study's contribution. Also, is there a well-established hypothesis connecting moisture heat transport to (concurrent) heat extremes in Rossby wave anomalies? Or are the authors proposing or testing a new mechanism? This lack of clarity makes it hard to follow how the study's "process-based investigation" (l.45) advances our understanding of co-occurring hemispheric heatwaves. Formulating specific research questions would go a great way to improve the clarity and goal of the paper.

*Thank you for the detailed criticism about the positioning of the work. This gives us the opportunity to specify what are the motivations behind this analysis, in the context of existing literature (most of which has been mentioned in the introduction):*

- *zonally integrated meridional heat transport extremes are related with the excitation of planetary-scale waves in both seasons (Lembo et al. 2019);*
- *heatwaves in the Northern Hemisphere are often associated with concurrent events spanning the most part of the mid-latitudinal channel (e.g. van Loon and Thomson, 2023);*
- *a clear and robust identification of heat transport extremes would allow to provide a coherent interpretation of how heat transports are related to concurrent heatwaves through the action of planetary-scale waves;*
- *the mechanism underlying this relation can be understood through the knowledge of changes in the dominant weather regimes that was discussed in a previous companion paper (Lembo et al. 2022);*

*Given these premises, while existing literature has mainly focused on the dynamical drivers behind the occurrence of hemispheric heatwaves, we want to highlight how meridional heat transport extremes link with (some of) these types of heatwaves through the excitation or dampening of specific planetary-scale waves. This relates the thermodynamic background energy flows to the occurrence of such events. We did not aim to provide any hypothesis about the role of moisture, as we think it falls beyond the scope of the present work.*

*We will substantially revisit the Introduction, addressing this suggestion and comments from the other reviewer.*

2. Regarding Section 4

   The explanation of the proposed mechanism is introduced only in the results section, and primarily through a case study. While I understand the use of a case study as a narrative device, in this context it would be more effective to introduce the mechanism earlier – ideally in the introduction – to better frame the study and guide the reader through the analysis. This could mean swapping the position of Sections 4 and 5, so that first we are presented with the overall picture, and then an example to show what a typical (or not typical) case might look like.

*Agreed. We will swap sections 4 and 5 (and the arguments in section 6) to reflect whit suggestion.*

   This section contains detailed descriptions that can be difficult to follow, but it remains unclear what they amount to. Adding a schematic (possibly even at the expense of Fig. 2) summarising the

key processes, stages, and mechanisms – whether in the case study or the general explanation, whichever helps improve readability – would greatly help readers grasp the overall picture more easily. One schematic for the summer case and another later for the winter case.

***Agreed. We will provide a depictive scheme in the Discussion section, trying to explain what is the proposed mechanism underlying the relation between MHT extremes and hemispheric heatwaves.***

3. As it stands, Sections 4–6 are very description-heavy, referring to multiple subfigures, variables, and seasonal differences. Because the results and discussion are presented as separate sections, it becomes difficult to digest and interpret the findings. A more integrated structure – e.g. combining results & discussion – could improve readability and help contextualise the findings as they are introduced. The schematic illustration suggested above could also aid in guiding the reader through the narrative.
4. Another possible restructuring tactic would be to subsume the current Sections 3–6 into a single Results section, with subsections (e.g. 3.1–3.4) capturing the existing material.

***We agree with the reviewer that the Results section could be substantially improved, making it more integrated to the Discussions section. We prefer addressing this by taking on point 4. suggestion, as this makes it easier to place the case studies in the context of the analysis.***

5. Towards the end of the study, authors suggest that the findings could help improve the predictability of certain events, it is currently not clear how this connection is established or supported by the results. That said, this might become more self-evident once the structure and contextualisation of the manuscript have been revised.

***Apologize for the confusion. We did not mean to imply that our results facilitate better predicability of hemispheric heatwaves. We refer to "tipicality" at the end of section 7 to emphasize that some conditional occurrences are somehow typical and relate (cfr. Lembo et al. 2022) to dominant weather regimes. We cannot infer how predictable this occurrences are given some values of MHT. We will state it more clear how this related to the typicality of heatwaves in the revised manuscript.***

**Specific comments**

l.5 "other conditional occurrences" is too abstract and uninformative – what are these other occurrences?

*We simply refer here to other combinations of LST and MHT extremes (e.g. cold LST-weak MHT, warm LST-strong MHT...). We will revise the phrase to clarify this.*

l.17 and 24 The use of 'evidence' as a verb is technically correct but sounds unusual in contemporary academic English and may read as non-idiomatic. A more standard phrasing such as 'shows' or 'highlights' or 'provides evidence for' would improve readability.

*Agreed. We will try to avoid using the verb "evidence" as much as possible.*

l.38 "underlying conditions" is vague here, as it appears to group together two quite different types of factors: fixed geographic characteristics (orography and land-sea contrast) and low-frequency or slowly varying boundary conditions (sea surface temperature anomalies). The phrase could be amended to be more precise.

*Agreed. We will revise the phrase in order to be more specific.*

ll.51-6 See general comment recommendations for re-structuring.

l.80 Numbering missing for equations pertaining to wavenumber decomposition.

*Apologize for the inconvenience. We will make sure that the equations are correctly labeled.*

l.93 add space before 'with'

*Agreed*

l.96 add space after *index*"

*Agreed.*

Sect. 2.2.2. Some key details are missing or unclear in this paragraph for full reproducibility. Detrending and deseasonalisation are mentioned, but the specific methods used aren't explained. It's also unclear how the declustering is done – what parameters or window are used? Finally, more detail is needed on how the EVT-based convergence algorithm works in practice.

*Agreed. Some details of the procedure are not explicit, as also outlined by a community comment. We will revise the manuscript in order to account for the missing explanation on the methodology.*

l.111 This sentence implies a directional, possibly causal relationship. I suggest using "association" instead of impact if the paper only shows statistical associations, and no causal analysis. If authors had introduced a clear hypothesised mechanism, then using terms like "impact" might feel more justified – even if causality isn't fully proven.

*Agreed. The sentence can read as a bold statement about causality, although it is not. This point was also raised by the other reviewer. We will revise the text in order to reflect that we are mainly describing statistical associations that we aim to interpret.*

l.114 "composite mean of such quantities in coincidence of LST|MHT" is a bit wordy – could simply write "composite mean of LST|MHT"

*Agreed. We will revise the text accordingly.*

ll.111-114 "…whether the composite mean … is random or reflects an emerging pattern." Is somewhat vague. What is the null hypothesis? i.e. are co-occurring MHT extremes associated with significantly different conditions *during* LST extremes?

*The null hypothesis is that the set of conditional occurrences is a random set of occurrences with fixed size, and that the anomalies thereby shown are non significant.*

l.155 what counts as a "consistent deviation"?

*Agreed. We should refer to it as "significant". We will revise the text accordingly.*

l.118 Would suggest adding a reference to Wilks (2016).

*Thanks. We will add the suggested reference.*

ll.119-20 Significantly different the OVERALL distribution? So are you testing LST|MHT against LST|‾MHT‾ or climatology? Please clarify.

*By "overall distribution" we refer here to "climatology" for what concerns masks applied in Figure 1 and B1, LST95 events for what concerns Figure 4 and Figure 6.*

***Apologize for not stating it clearly here. We will revise the manuscript in order to be more precise.***

l.122 "weak MHT extreme" sounds contradictory. If the intended meaning is "extremely weak MHT values (i.e. very low MHT, and not non-extreme or moderate), then a clearer phrasing would be "extremely weak MHT" or possibly "strongly negative MHT anomalies" or "lower-tail MHT extremes." I suggest considering one of these alternatives *throughout the manuscript*.

***Agreed. We used the term "extremely weak MHT" in several instances, so we would stick with this terminology, whenever we do not use the related symbol.***

Table 2. Last row highlights values below 10%, in contradiction with caption.

***Thank you for spotting the inconsistency, that was also detected by reviewer 1. We will correct the caption accordingly.***

ll.126-9 What is the implication for your study of the statement in the sentence starting "However, [...]" with regard to the strength of the association between MHT -> LST ?

***We carried out the same analysis by both including and excluding <5 days lasting heatwaves, but we did not notice significant qualitative differences. Therefore, in order to preserve a larger number of instances, and for consistency between boreal Summer and Winter, we preferred maintaining all heatwaves with any duration. We will revise the analysis on only >5days and we may include this as an additional Appendix, in case we would notice that there would be anything worth highlighting.***

l.128 "consecutive days" is more standard phrasing in this context.

***Agreed. We will revise the sentence accordingly.***

ll.130,133 "weak MHT-warm LST" is the same as $LST_{95}|MHT_{\downarrow}$, correct? I suggest sticking with the latter 'symbolic' notation throughout the text for clarity and consistency. Reverting to mixed phrases like the former can easily lead to confusion, especially when other conditions (like strong MHT) are also discussed.

***Agreed. We will try to use the symbolic notation everytime it is possible.***

ll.130-33 One of the study's key claims regards $LST_{95}|MHT_{\downarrow}$ accounts for 30% and 40% of warm LST days in JJA and DJF, respectively. However, this paragraph suggests

that these figures come from a fraction of heatwaves, whose total durations make up that proportion of LST days. If so, I think this deserves clearer framing – the current phrasing risks implying that individual $LST_{95}|MHT_{\downarrow}$ days make up that share, rather than those days occurring within longer events. I recommend clarifying whether the percentages refer to discrete days or to the cumulative duration of events in which $LST_{95}|MHT_{\downarrow}$ conditions are observed. This is a presentation issue, not necessarily a flaw in the result, but it risks misleading interpretation.

***Indeed, we claim that the conditioned events are related to 30.4% >5days events in DJF, 42.5% in JJA. This is not meant to say that all days belonging to the heatwave are characterized by a conditional occurrence. We will state it very clearly at the beginning of this paragraph and elsewhere in the text, where relevant.***

Figure 1. I suggest reordering the figure panels so that the same variable is shown side by side across seasons. This would make seasonal differences easier to compare at a glance.

***This is a very good point. Thank you for the suggestion. We will certainly implement it.***

l.156 The interpretation of wind speed anomalies is not straightforward when trying to infer changes in the jet stream's absolute strength or position. A positive anomaly simply indicates stronger-than-average winds at a given location, but this can mean very different things depending on the climatological mean – e.g. in climatologically weak-jet regions, it may reflect modest flow or jet extension, while in strong-jet regions, it could signal true intensification or a shift in jet position. I suggest supplementing the V250 fields with contours of absolute wind fields or some baseline seasonal climatology.

***Agreed. A similar suggestion was also provided by reviewer 1 and will certainly find the best way to implement it. We may have to use a different strategy than what proposed, given that the contours could be shaded by the significance masks, but we will address it fully before submitting the revised manuscript.***

ll.177-8 The authors attribute the pattern to strong blocking based on SLP anomalies. However, since blocking is generally diagnosed using mid- to upper-tropospheric geopotential height or potential vorticity. I suggest either supporting this claim with such fields or rephrasing to avoid implying a definitive blocking event based on surface pressure alone.

*Agreed. Thank you for raising this point. As we acknowledge that the blocking is usually computed from geopotential height fields at 500 hPa (cfr. Davini et al. 2012), we should refer to these patterns more properly as high/low pressure systems. Nevertheless, as this point was raised by reviewer 1 as well, we realise that the manuscript would benefit from a quantitative estimate of blocking index over the usually active regions, we will provide this, based on the Tibaldi-Molteni index and successive updates (e.g. Davini et al. 2012) and compare this with what can be drawn out of considerations about SLP fields, especially in Summer (cfr. Katsafados et al. 2014).*

ll.186-7 See my comment regarding l.156.

*Agreed. We will make the climatological fields explicit in the revised figures and text.*

Figure 3. I recommend increasing the font size of all text and labels, as well as increasing the thickness of curves in Fig. 3b to improve visual communication. Additionally, Fig. 3a title says MHT anomalies, but the caption says absolute MHT. Please clarify.

*Agreed. We will increase the size of axis labels, the thickness of lines in Figure 3b, and replace the title of panel 3a with the correct one. Apologize for the lack of visual clarity.*

l.219 Are authors making this claim based on the one case?

*This point was raised by Reviewer 1 as well, therefore we apologize for not making it clear enough that we hereby refer to the results from White et al. 2022 regarding the 2021 NW Pacific heatwave, and we mean here to confirm that this argument can be generalized to this class of cases for boreal Summer. We will revisit the phrase accordingly.*

l.258 Double-check WCD's stylistic guide for dates: I think "19 January 2007" without 'th' may be correct.

*Sure. This is a typo and we will certainly correct that.*

l.274 "O" letter instead of "0" number in ENSO

*Thanks for spotting it. We will revisit the text accordingly.*

ll.299-300 What are the implications of these observed differences? Clarifying this sentence would strengthen the connection between the diagnostics and the study's core questions – which, as currently presented in the introduction, are not that clear and could benefit from sharper formulation.

*Thank you for this comment. We realise that we should dig a bit further in the interpretation of these differences. Therefore, we will expand this paragraph and be more specific about how this is relevant for our understanding of MHT extremes role for hemispheric heatwaves.*

ll.322-6 This is precisely the kind of information that could be part of a summarising schematic.

*We think that this summarizing sentence is useful in this context, in order to avoid the reader being confused by the chain of arguments above. We will make sure to repeat or at least mention more explicitly this concluding remark in the Conclusions.*

ll.328-31 An interesting hypothesis that would be worthwhile testing in future research!

*Thanks! We do in fact plan to investigate these connections in future climate change scenarios across CMIP-class models.*

l.345 The phrase "other combinations" is vague and uninformative. I recommend briefly restating the specific combinations for clarity, even if this involves some repetition.

*Agreed. We will try to recall the conditional occurrences combinations briefly at the beginning of the Conclusions.*

l.348 Once again, I would hesitate to mention causality.

*Agreed. This hints once again at the necessity of clarifying the aims and scope of this work, that will be among the main efforts required to revisit the manuscript and prepare a new version of it. We take full responsibility for this lack of clarity and plan to convey the relevant message in a more explicit way in the conclusions.*

Figure 7b and various in panels Figure 9. Some anomalies exceed the colorbar range, resulting in blank or clipped areas in the plots.

*That is true. We were a bit hesitant in changing the colorbar range, in order to maintain a range that is coherent across different figures and avoid shading the relevant results. We will revisit the best combination of colorbar range that allows to convey the message and fit in the whole range of anomalies in each panel.*